# Roles of SNORD115 and SNORD116 ncRNA clusters during neuronal differentiation

Aleksandra Helwak [1] ✉, Tomasz Turowski [1,2], Christos Spanos [1] & David Tollervey [1] ✉

In the snoRNA host gene *SNHG14*, 29 consecutive introns each generate SNORD116, and 48 tandem introns encode SNORD115. Loss of SNORD116 expression, but not of SNORD115, is linked to the neurodevelopmental disease Prader-Willi syndrome. SNORD116 and SNORD115 resemble box C/D small nucleolar RNAs (snoRNAs) but lack known targets. Both were strongly accumulated during neuronal differentiation, but with distinct mechanisms: Increased host-gene expression for SNORD115 and apparent stabilization for SNORD116. For functional characterization we created cell lines specifically lacking the expressed, paternally inherited, SNORD115 or SNORD116 cluster. Analyses during neuronal development indicates changes in RNA stability and protein synthesis. These data suggest that the loss of SNORD116 enhances some aspects of developmental timing of neuronal cells. Altered mRNAs include *MAGEL2*, causal in the PWS-like disorder Schaaf-Yang syndrome. Comparison of SNORD115 and SNORD116 mutants identifies small numbers of altered mRNAs and ncRNAs. These are enriched for functions potentially linked to PWS phenotypes and include protocadherins, which are key cell signalling factors during neurodevelopment.

Prader-Willi syndrome (PWS) is a paradigm of neurodevelopmental disorders with a frequency of ~1:20,000[1–4] caused by lack of expression of genes in chromosome 15, region 15q11.2-q13 (Fig. 1). The region is imprinted with different expression patterns for the paternal and maternal chromosomes. PWS specifically concerns the expression of genes from the paternal chromosome due to deletion, uniparental disomy, or imprinting center defects. Deletions in the maternal chromosome cause a different neurological disease, Angelmann syndrome. This has been linked to the loss of *UBE3A*, which encodes a ubiquitin ligase[5] (and references therein). Deletions causing PWS typically remove large genomic regions, however, disease-linked microdeletions have also been identified[3,6,7]. The smallest remove a ~71 Kb region of *SNHG14* (snoRNA host gene 14), in which 29 tandem introns each encode the small nucleolar RNA (snoRNA) SNORD116 (Fig. 1) and the poorly characterized ncRNA *IPW*[1,6–8].

*SNHG14* generates a very long non-protein coding RNA (lncRNA) with a predicted primary transcript ~600 Kb in length, including 145 annotated introns[9]. It is processed into multiple overlapping ncRNAs; including mature snoRNAs, extended snoRNA-related ncRNA species (SPA-lncRNAs and sno-lncRNAs), and alternatively spliced versions of the *SNHG14* exons[10,11] (reviewed in ref. 12). Multiple, non-identical versions of SNORD116, are encoded by 29 tandem introns of *SNHG14* and excised following splicing. Adjacent to the SNORD116 region, a further 48 tandem introns encode another snoRNA-like species, SNORD115. However, differences in the spatiotemporal expression profiles of SNORD115 and SNORD116 have been reported[13,14], potentially reflecting a boundary conferring tissue-specific expression of the *SNORD115* and *UBE3A-ATS* regions and involvement of transcriptional activator CTCF[15,16]. Deletion of the SNORD115 cluster alone does not result in human PWS[17].

PWS individuals show a range of developmental and neurological deficits. Perhaps most notable is hyperphagia, which leads to potentially life-threatening over-eating. This has been linked to altered gene expression in the hypothalamus, where hunger is regulated[4,18,19].

[1]Institute for Cell Biology, School of Biological Sciences, The University of Edinburgh, Edinburgh, Scotland. [2]Institute of Biochemistry and Biophysics PAS, Warszawa, Poland. ✉e-mail: ahelwak@ed.ac.uk; d.tollervey@ed.ac.uk

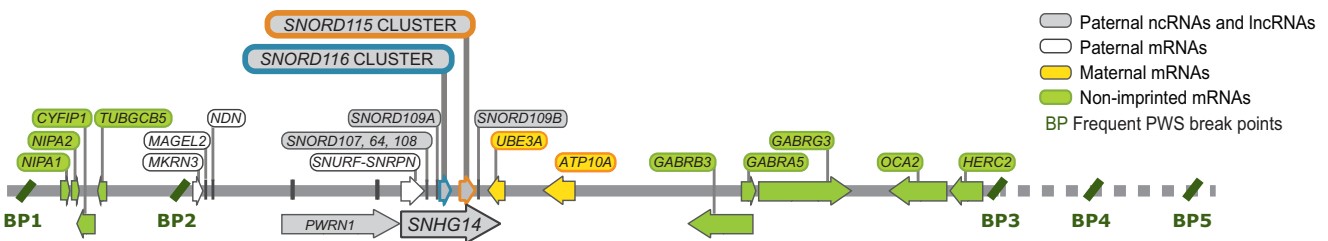

**Fig. 1 | Schematic showing the structure of the PWS locus and the transcription units within this region.**

Human tissue distribution data confirmed high SNORD116 levels in multiple brain regions including, but not limited to, the hypothalamus. These findings suggest direct roles for the snoRNA in gene expression in pathways regulating feeding, which could be partially reproduced in mouse models (reviewed in ref. 20).

Human snoRNAs are frequently encoded within introns of mRNAs or long non-protein coding RNAs. In most analyzed cases, the mature snoRNAs are generated by 5′ and 3′ exonuclease degradation of the excised intron following splicing and debranching. Progression of the exonucleases is likely blocked by snoRNA assembly with proteins into stable, small nucleolar ribonucleoprotein (snoRNP) particles since the loss of these proteins in yeast prevents snoRNA accumulation[21].

SNORD116 and SNORD115 species resemble box C/D class snoR-NAs, which have characteristic structural features and bind a set of four, highly conserved proteins; methyltransferase Fibrillarin (FBL), NOP56, NOP58, and SNU13, (NPHX, 15.5 K). All characterized snoRNAs function through base-pairing with target RNAs, most commonly directing site-specific modification in rRNAs or other small stable ncRNAs, while some are required for pre-rRNA processing. Box C/D snoRNAs generally form extended base-paired interactions that precisely target the 2′-hyrodroxyl residue on the nucleotide located 5 base-pairs from the box D motif in the snoRNA. This directs nucleotide-specific 2′-O-methylation of the ribose group by the snoRNA-associated methyltransferase FBL[21]. This specificity allows target sites for many snoRNAs in rRNA and stable ncRNAs to be precited with considerable confidence. However, no relevant targets are known for SNORD116[8]. Potential targets in mRNAs have been predicted[22], and reporter constructs indicate that ectopic SNORD116 expression can stabilize the *NHLH2* mRNA[23,24].

Most snoRNAs are ubiquitously expressed, but ~200, including SNORD116 and SNORD115 were reported to show brain-enriched expression[8]. Most of these are described as orphans since, like SNORD116 and SNORD115, they lack evident base complementarity to rRNA or other targets. Loss of SNORD115 was previously reported to impair specific pre-mRNA splicing and editing events, including that encoding neuronal serotonin receptor 2 C (HTR2C)[25,26], but the significance remains unclear[27,28]. Changes in mRNA levels have been reported in human PWS-derived cells, brain samples, and SH-SY5Y cells, but with limited consistency[4,22,23,29] (reviewed in ref. 30). In addition, extended forms of the snoRNAs have been proposed to sequester specific RNA-binding proteins, including the pre-mRNA spicing factor RBFOX2[11,31]. The *SNHG14* host gene is widely expressed, with brain enrichment (reviewed in ref. 12). Despite these findings, the mechanistic basis for links between the non-coding RNAs originating from the *SNHG14* locus and PWS remains unclear.

In a neuronal model, we tested mechanisms by which RNAs encoded by SNORD115 and SNORD116 clusters might alter gene expression. Partially overlapping changes in RNA abundance and predicted translation were observed on the loss of either snoRNA. Since the absence of SNORD116 but not SNORD115 has been linked to PWS, we focused on changes specifically shown in the absence of SNORD116. This generated a small list of genes, with clear enrichment for functional associations with PWS.

## Results

### Regulated expression of ncRNAs from the *SNHG14* locus

Consistent with their reported brain-enriched expression[8] we saw no SNORD115 or SNORD116 in HEK293 cells. However, we detected expression in human Lund human mesencephalic (LUHMES) cells, an embryonic, mid-brain derived human cell line, that can be induced to synchronously differentiate into polarized dopaminergic neurons[32,33] (Fig. S1A). High synchrony was of particular importance for biochemical analyses during differentiation time courses. Neuronal differentiation can be followed in long-term studies[34,35], with multiple neuronal markers expressed by day 6 of differentiation[32,33]. We tested snoRNA expression over differentiation time courses up to day 15. In cycling, pre-neuronal cells (day 0; D00) SNORD116 was readily detected by northern hybridization, whereas SNORD115 was not detectable (Fig. 2A, S1B). During differentiation, SNORD116 abundance increased, reaching a plateau between day 6 (D06) and day 10 (D10), as judged by northern hybridization (Figs. 2A and S1C). RT-PCR estimated ~fivefold increase between D00 and D15 (Fig. S1B). In contrast, SNORD115 was first detected at D06 and steadily increased up to day 15 (D15) based on northern hybridization, with a much weaker signal than SNORD116 at all time points. The relatively late accumulation of SNORD115 and SNORD116 suggested their importance during later stages of neuronal maturation.

RNA sequencing (RNA-seq) was performed on undifferentiated cells and during differentiation at D06, D10, and D15 (list of samples in Supplementary Data 1). RNA-seq data were initially characterized for expression of ncRNAs originating from *SNHG14* (Fig. 2B). *SNHG14* ncRNA abundance over the region surrounding the *SNORD116* cluster was almost unaltered during differentiation. Accumulation of the exons was substantially greater than for introns, which are normally rapidly degraded following debranching, consistent with correct splicing of introns encoding SNORD116 throughout differentiation. Supporting this conclusion, northern analysis failed to detect unprocessed introns at any time during differentiation (Fig. S1C). In contrast, the region surrounding the *SNORD115* cluster showed low expression at D00. This increased during differentiation but remained substantially lower than around *SNORD116* even at D15. To better characterize expression changes, reads mapping to exon sequences were compared (Fig. 2C). This confirmed that transcripts around *SNORD115* markedly increased during differentiation, while those around *SNORD116* were essentially unchanged. Exons overlapping *SNUFR-SNRPN* showed increased levels and contributed most to the overall increase in *SNHG14* abundance (Fig. 2B). Extended snoRNA-related RNAs SPA1 and sno-lncRNAs 1–4[10,11,31] were readily detected (Fig. 2B) but showed only modest changes during differentiation. Other reported ncRNAs, SPA2 and sno-lncRNA 5, were not clearly identified. Northern analysis (Fig. S1C) showed that mature SNORD116 accumulates at much higher levels than extended, snoRNA-related ncRNAs throughout differentiation.

Previous analyses of *SNHG14* expression reported a single primary transcript[17] or independent transcription units across *SNORD115* and *SNORD116*[13] in humans, and inclusion of multiple upstream exons in mice[14]. Our sequence data showed a clear drop between the *SNORD116*

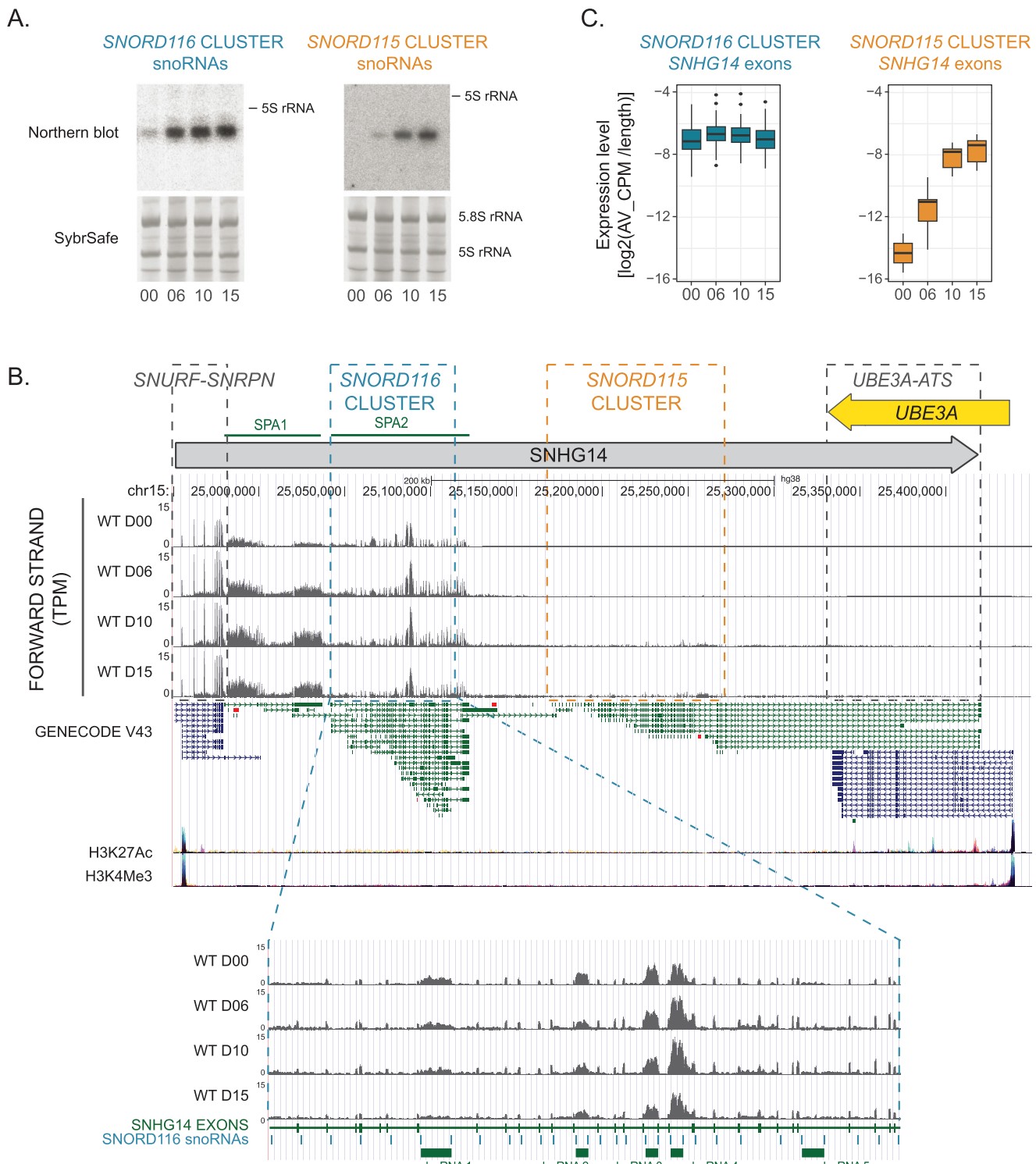

**Fig. 2 | Changes in expression of ncRNAs from *SNHG14* gene during neuronal differentiation. A** Expression of SNORD115 and SNORD116 snoRNAs in LUHMES cells upon differentiation. Northern blot and corresponding SybrSafe stained fragment of the gel below. The approximate position of 5S rRNA (120 nt) is marked on the Northern blot. Similar profiles of SNORD115 and SNORD116 expression were obtained at least three times. **B** UCSC Genome Browser view of the expression from the *SNHG14* gene at day 0, 6, 10, and 15 of differentiation. GENECODE 43 track displays a basic gene set with splice variants, mRNAs (blue), and non-coding RNAs (green). Layered H3K27Ac and H3K4Me3 tracks display data on histone modifications from ENCODE project, associated with the enhancer/regulatory regions and promoters, correspondingly. *SNORD116* and *SNORD115* cluster regions within the *SNHG14* gene are marked with boxes. SPA and sno-lncRNAs ncRNAs, previously described but not included in the GENE-CODE track, are marked at the top and bottom of the figure. **C** Difference in expression profiles of *SNHG14* lncRNA exons in RNA-seq data, overlapping either *SNORD116* or *SNORD115* cluster. Expression levels (CPM) for individual exons are length normalized. The distribution of expression levels for each timepoint is based on $N = 4$ (days 00, 06, 10) or $N = 3$ (day 15) independent biological replicates. The center line represents the median, bounds of box–lower and upper quartile; the whiskers are reaching the largest or the smallest value, at most 1.5 IQR of the bounds, outliers are marked with the dots. Source data are provided as a Source Data file.

and *SNORD115* clusters. To investigate this, we inspected mapping data for histone H3 lysine 4 trimethylation (H3K4me3), characteristic of RNAPII transcription initiation sites, and H3K27Ac, characteristic of regulatory regions (see; www.ncbi.nlm.nih.gov/gene/104472715 and UCSC genome browser ENCODE regulation tracks)[36]. H3K4me3 peaks, and accompanying H3K27Ac peaks, were found at the predicted transcription start sites for *SNHG14* and the flanking *UBE3A* protein coding gene (Fig. 2B). There was no indication of initiation between the *SNORD116* and *SNORD115* clusters. We, therefore, predict that the apparent extension of the *SNHG14* transcripts into the region surrounding the *SNORD115* cluster reflects a regulated readthrough of a termination site located 3′ to *SNORD116*. Consistent with this, GENE-CODE V43 (Fig. 2B) indicates the processing of *SNHG14* transcripts into multiple alternatively spliced versions (in non-neuronal cells), covering *SNORD116* or *SNORD115* but not overlapping both clusters.

We conclude that the region encoding SNORD116 is well transcribed in undifferentiated cells. The accumulation of exon regions relative to introns indicates that splicing is functional, suggesting that low accumulation of mature SNORD116 reflects instability. We speculate that impaired assembly with snoRNP proteins allows degradation of the snoRNA sequence along with the excised intron within which it is embedded. At the same time, there is a clear expression of SNORD116-containing sno-lncRNAs, indicating that these two types of ncRNAs undergo distinct, possibly competing processing pathways. In contrast, SNORD115 is poorly expressed in undifferentiated cells, with increased transcription readthrough into this region during differentiation.

## Transcriptional changes during differentiation of LUHMES cells

LUHMES cells are frequently used as a model for dopaminergic neurons. Changes in the transcriptome[33] and proteome[37] were analyzed up to day 6 of neuronal differentiation, at which point they were considered mature. In agreement with that, in our data, most changes to the transcriptome occur between D00 and D06 of LUHMES differentiation (Fig. 3A, C). From all genes exhibiting altered expression during differentiation in the wildtype, only about 1% showed changes between D10 and D15, when snoRNA-linked effects were most common (see below).

To identify gene expression patterns related to neuronal differentiation, we performed k-means clustering, with $k = 6$ giving the best resolution without repeating patterns (Fig. 3B, C). Principal component analysis (PCA) for all quantified genes, confirmed good separation between clusters (Fig. 3B; genes are colored by cluster). The results are consistent with previous data on LUHMES cells and other analyses of neurogenesis[33]. The biggest cluster, CL0 (25.2% of all genes) contains genes that are stably expressed throughout differentiation. GO term analysis using gProfiler, indicates enrichment for intercellular transport, transcription, proteolysis, and macro-autophagy; essential functions regardless of differentiation status (Fig. 3D; detailed results in Supplementary Data 2). CL1-3 comprise genes that substantially change expression between D00 and D06, as the LUHMES cells exit mitosis. CL1 and CL2 genes showed decreased expression (CL2 more acutely than CL1) and are involved in growth, cell cycle regulation and progression, transcription, ribosome biogenesis, and translation. CL3 genes had increased expression at D06 but lacked clear GO term enrichment. The smaller CL4 (11%) and CL5 (6%) comprise genes that continued to change later during the differentiation time course. Genes from CL4 generally showed elevated expression at D06 that continued over later time points. They are enriched for characteristic neuronal functions, including neurite development, axon guidance, intercellular communication, formation of synapses, cell junctions, transmembrane transport, or cell motility. CL5 genes rose sharply at D06 and then declined. They show enrichment for tissue development and establishment of higher-level organization, particularly muscles–both striated and cardiac. Decreased expression after an initial peak, potentially indicates

similarities in early developmental pathways for neurons and muscles, with divergence at this point.

Notably, genes from clusters CL4 and CL5 are enriched for neuronal/developmental functions (Fig. 3D) and show regulated expression at times when SNORD115 and SNORD116 snoRNAs accumulate.

## RNA abundance changes in a disease model system

To understand the roles of *SNORD115* and *SNORD116* in neuronal gene expression, we precisely deleted regions of *SNHG14* containing the snoRNA clusters from only the paternal (expressed) chromosome, using CRISPR in LUHMES cells (Fig. S2A). The heterozygous deletion strains were designated H115 (1 clone) or H116 (2 independent clones), respectively (see Material and Methods). A homozygous *SNORD115* deletion clone was initially included in the analysis, but showed a gene expression phenotype distinct from any single mutant line, in some cases being more similar to the wildtype. The basis of this is unclear, but it was excluded from further analyses. LUHMES cells are reported to be diploid[38] with normal karyotyping. Analysis by PCR confirmed the heterozygous deletion (Fig. S2A), and northern hybridization demonstrated the expected absence of snoRNA expression in H115 and H116 (Fig. S2B), thus confirming paternal deletion and integrity of the maternal chromosome.

The heterozygous deletion strains did not exhibit clear changes in neuronal morphology, and RNA-seq data (below) did not show increased expression of genes linked to cell death (*ANXA5*, *CASP3*, *CASP7*, *TP53*) (Fig. S3B), indicating that cell viability is not substantially impaired.

Most canonical box C/D snoRNAs function as modification guides for stable RNA species[7], predominantly rRNAs, but a minority are required for correct pre-rRNA processing. Complementarity between SNORD115 nor SNORD116 and the pre-rRNA was found, but neither showed the very specific interaction pattern expected to be required for methylation[8]. The interactions of the few snoRNAs that promote pre-rRNA folding and processing are more heterogenous. However, comparison of pre-rRNA processing in the wildtype, H115, and H116 strains by northern hybridization (Fig. S2C) revealed no clear differences.

To assess the effects of the deletion of *SNORD115* and *SNORD116* clusters on general gene expression, we performed RNA sequencing at time points from D00 to D15, using a bulk, non-nucleofected population of WT LUHMES cells without clonal selection as a negative control. Due to their short lengths, mature snoRNAs were not detected. RNA-seq analysis confirmed the accurate deletion of the entire *SNORD116* region from H116 and the *SNORD115* region from H115 (Fig. 4A). Total numbers of reads mapping to *SNHG14* were reduced consistent with the deleted region (Fig. 4C), indicating that transcription per se was not affected. Expression of genes in proximity to the *SNORD116/115* clusters was unaffected, including *UBE3A*, which is adjacent to *SNHG14* but transcribed only from the maternal chromosome (Fig. 4C). In the wider *PWS* locus region (Fig. 4B, D), we observed under-accumulation of *MAGEL2* mRNA at D10 and D15 in both deletion strains, with a greater effect in H116. The *MAGEL2* gene, which is causal for Schaaf-Yang syndrome, is transcribed on the opposite strand from *SNHG14* and located around 1.5 Mb upstream. In addition, both mutant cell lines showed reduced expression of *OCA2*, and increased *GABRA5* expression.

Transcriptome-wide, most mRNAs that changed expression during wildtype differentiation showed similar changes in H115 and H116 (Figs. 5A and S3A). Good reproducibility was seen across four replicates for the wildtype and H116, and two for H115 (Fig. S3A). For selected mRNAs, the changes seen in RNA-seq were confirmed with RT-PCR using independent RNA preparations, along with one set of samples included in RNA-seq analyses (Fig. S3C). Across all stages, 731 transcripts were significantly altered between wildtype and H116 cells, with 411 transcripts altered between wildtype and H115 (Supplementary Data 3). The number of RNAs differentially accumulated

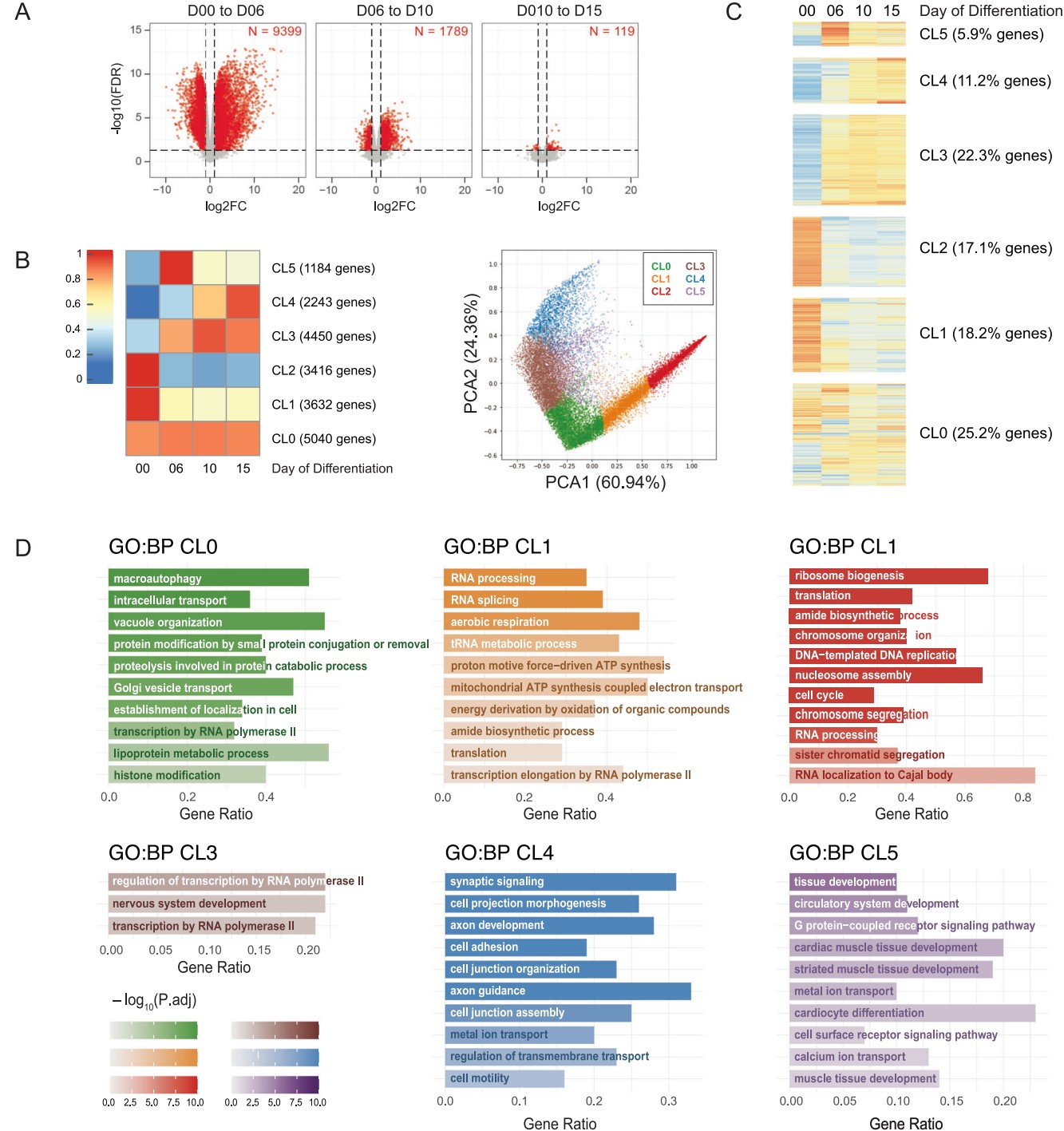

**Fig. 3 | Changes in gene expression during neuronal differentiation. A** Changes in gene expression associated with the differentiation process in wildtype LUHMES cells. Genes with statistically significant changes in expression between consecutive time points of differentiation as assessed by EdgeR (FDR; quasi-likelihood F-test followed by Benjamini-Hochberg correction for multitesting) are marked red. Source data are provided in Supplementary Data 3. **B** Clusters representing various gene expression patterns associated with the differentiation of LUHMES cells, obtained by k-means clustering of RNA-seq data. k-means of 6 resulted in a good representation of expression patterns and separation of genes into clusters, as seen in the PCA analysis plot on the right. Each point on the plot represents a gene, genes are colored by clusters. **C** Heatmap representation of expression patterns of all the genes during differentiation of WT cells, separated by clusters. **D** Representation of GO terms associated with defined gene clusters, as analyzed by g::Profiler using default statistical parameters: Fishers' one-tailed test and multiple testing correction algorithm g:SCS. Detailed outcome of the analysis is in Supplementary Data 2. Source data for the figure are provided as a Source Data file.

between mutant and WT cells lines was highest at D15 (Fig. 5B). This was notably later than most changes related to differentiation (Fig. 3A). Altered expression was seen for both mRNAs and lncRNAs, with a predominance of reduced expression (Fig. 5C). Hierarchical clustering (Fig. 5D) confirmed that at D00 and D06 the wildtype and

mutant cell lines cluster together, although the mutants are already more similar to each other. At D10 and D15, mutants cluster together, away from the wildtype. This reflects the considerable overlap between RNAs altered in H115 and H116 (295 transcripts) (Fig. 5E, F; and see Discussion).

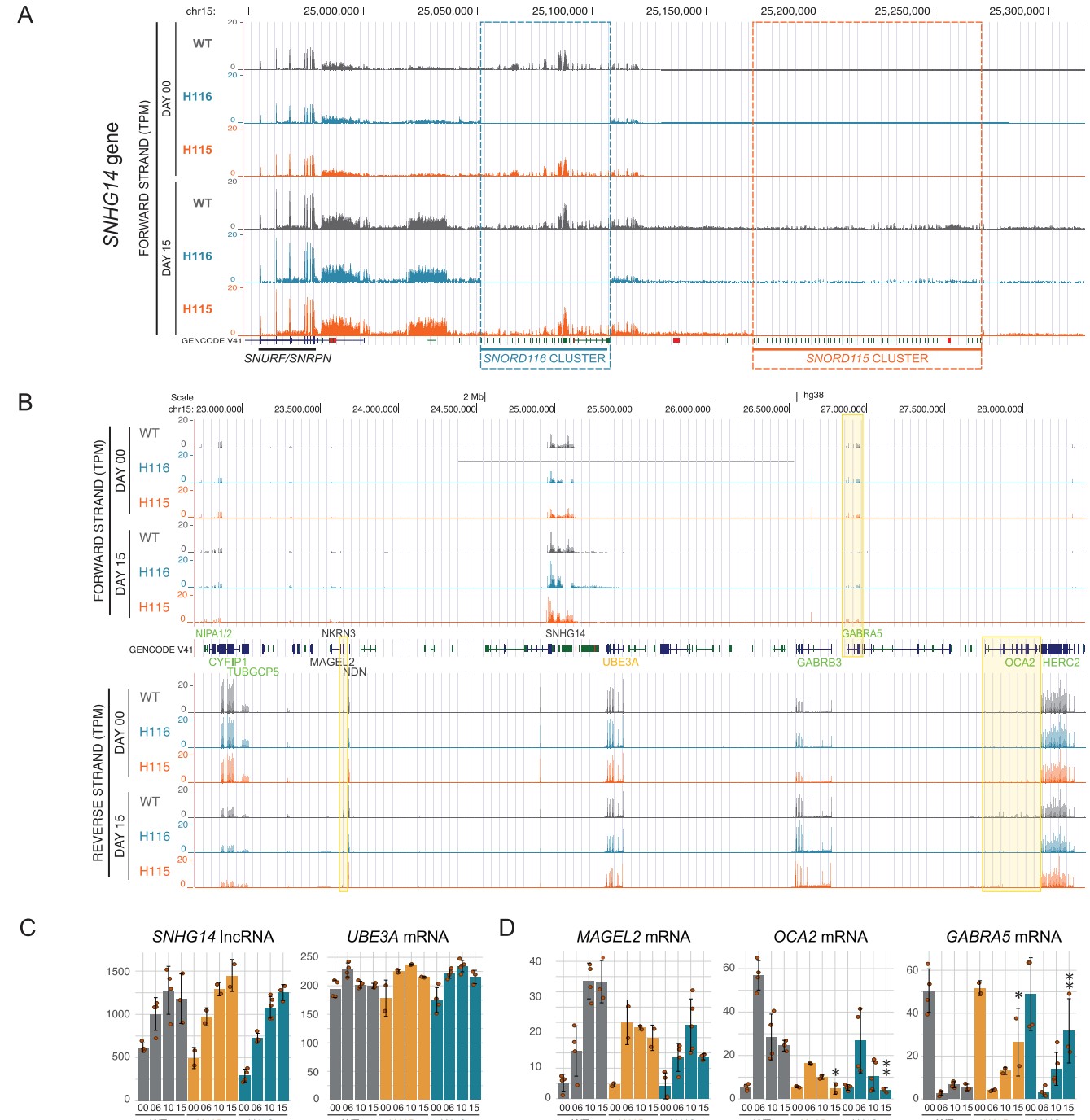

**Fig. 4 | Local effects of SNORD115 and SNORD116 cluster deletions. A** UCSC genome browser view of transcription across *SNHG14* gene in wild-type and mutant cell lines at D00 and D15 with indicated deletion regions. **B** UCSC genome browser view of transcription across *PWS* locus in wild-type and mutant cells. Marked are the genes with altered expression in the deletion mutants. Significance tested for mutant vs wildtype expression within the same differentiation stage using EdgeR (FDR; quasi-likelihood F-test followed by Benjamini-Hochberg correction for multitesting). **C** Mean expression of *SNHG14* lncRNA and *UBE3A*, convergent gene overlapping with *SNHG14*, during differentiation in wild-type and mutant cells. **D** Mean mRNA expression from *MAGEL2, GABRA5,* and *OCA2* genes from PWS locus is affected by the deletion of *SNORD115* and *SNORD116* clusters. **C, D** Marked statistical significance for the difference in expression between mutant and WT cell lines originates from EdgeR analysis (FDR; quasi-likelihood F-test followed by Benjamini–Hochberg correction for multitesting), error bars represent standard deviation (SD). The number of biological replicates for each mutant and timepoint is provided in Supplementary Data 1. Source data are provided as a Source Data file.

RNAs showing altered accumulation in H116 vs WT cells were compared to the 6 clusters defined from WT differentiation (Figs. 3B, C, S4A, B). DEGs in H116 were enriched in the two smallest clusters CL4 (30% of DEGs) and CL5 (29% of DEGs), which largely comprise neuronal-related genes regulated at later stages of differentiation. Downregulated genes were enriched in both CL4 and CL5, whereas upregulated genes were clearly enriched only in CL4. For the DEGs, GO term enrichment indicated processes characteristic of developing neuronal cells: regulation of membrane potential, axonogenesis, response to cAMP, endocrine system development; with highest enrichment for "cell surface receptor signaling pathway". Enriched terms in "Cellular Compartment" indicated association with membranes, secretion, and cell-cell junctions (Fig. 5G and Supplementary Data 4). Those terms agree with previous observations by Burnett et al.

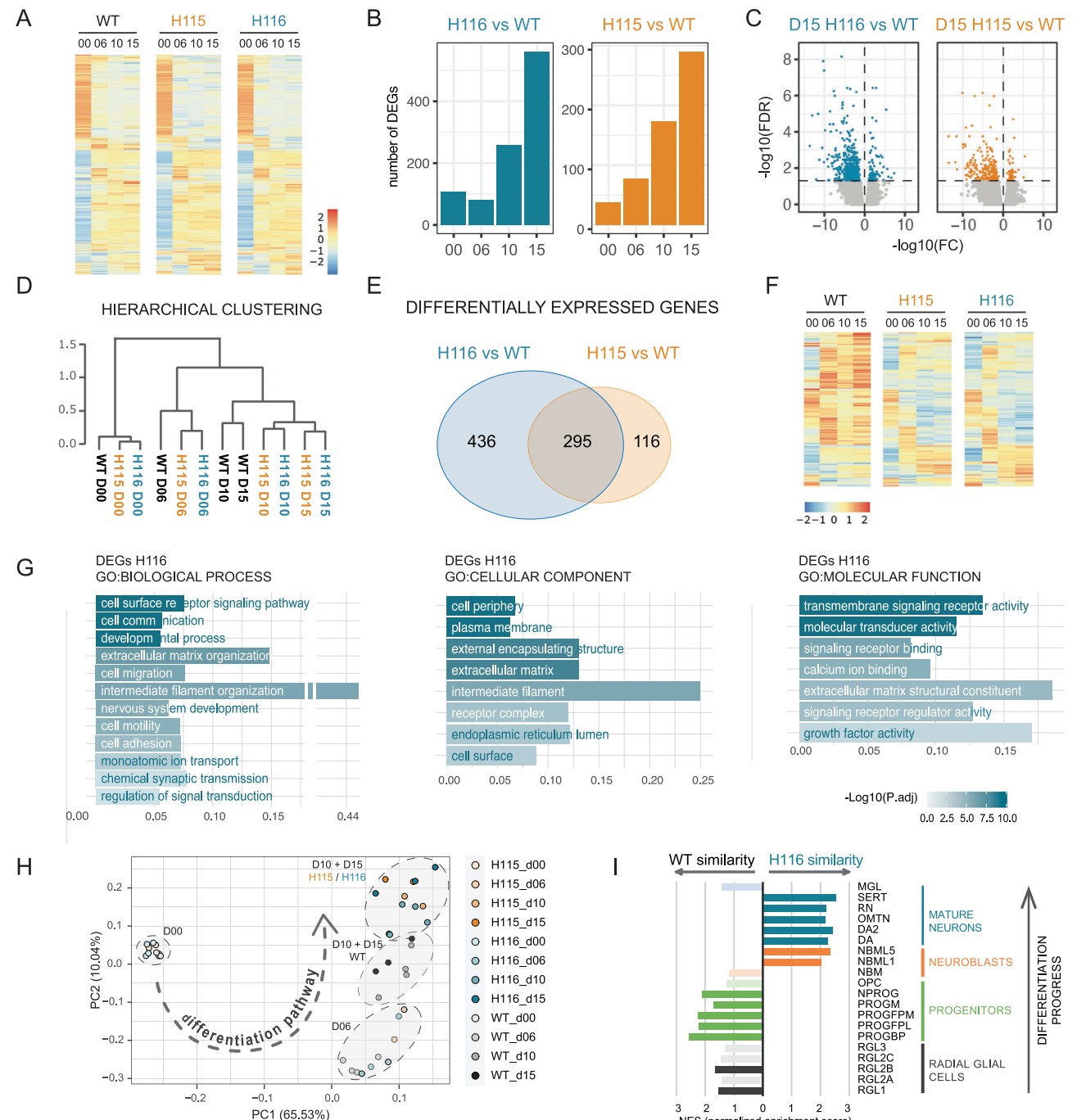

**Fig. 5 | Effects of *SNORD115* and *SNORD116* cluster deletions on transcriptome.**
**A** Heatmaps representing global changes to the transcriptome in differentiating wild-type and mutant cell lines. Mean expression (CPM) values are scaled for each gene across all the timepoints and all cell lines. **B** Number of differentially expressed genes between wildtype and mutant cells increases with the progress of differentiation. **C** Changes to gene expression in mutant vs wildtype cells at day 15 of differentiation, indicating a tendency towards decreased expression.
**D** Hierarchical clustering of RNA-seq samples shows high similarity between undifferentiated cells and subsequent separation between WT and deletion mutant cell lines. **E** Overlap between differentially expressed genes (DEGs) for H115 and H116 cell lines. **F** Gene expression profiles of DEGs, show very high similarity between H115 and H116 cell lines. Heatmap covers all 847 DEGs identified independently in both mutant cell lines and at all timepoints. **G** A summary of the GO

term enrichment analysis for H116 DEGs from g::Profiler, using default statistical parameters: Fishers' one-tailed test and multiple testing correction algorithm g:SCS. Full outcome of the analysis is in the Supplementary Data 4. **H** PCA analysis of RNA-seq data implies accelerated differentiation of H115 and H116 cells lines along the assumptive differentiation pathway. **I** Comparison of H116 and WT transcriptomes at D15 of differentiation with transcriptomes of various cell types identified during human mid-brain development (scRNA-seq)[41]. The mutant cells are more similar to mature neurons relative to the WT. GSEA analysis (see Methods). RGL1-3 radial glia-like cells, PROG progenitor cells, BP basal plate, FPL lateral floorplate, FPM medial floorplate, M midline, NPROG neuronal progenitors, OPC oligodendrocyte precursor cells, NBML1-5 mediolateral neuroblasts, DA-DA2 dopaminergic neurons, OMTN oculomotor and trochlear nucleus, RN red nucleus, SERT serotonergic, MGL microglia. Source data are provided as a Source Data file.

(2017) comparing iPSC-derived neurons from individuals with PWS and healthy controls, and Bochukova et al. (2018) based on the analysis of PWS hypothalamus (we do not observe pathways connected with inflammation in the analysis of our neuronal cultures).

## Altered gene expression timing in the absence of SNORD116

To better characterize the changes in gene expression that underlie separate clustering of WT, H115 and H116 samples, the transcriptome data was subjected to principal component analysis (Fig. 5H)[39]. In agreement with hierarchical clustering, initially similar cell lines gradually diverge during the differentiation process. By inspection, it appeared possible that the wild-type and mutant cells were on a similar trajectory of differentiation, but with greater progress in the mutants.

To explore this hypothesis, gene expression at D15 of differentiation in wildtype and H116 cells was compared to a large set of midbrain derived neuronal cell types. Gene Set Enrichment Analysis (GSEA)[40] was applied using published single-cell RNA sequencing data[41] as deposited in the Molecular Signatures Database (MSigDB)[40]. In Fig. 5I, cell types identified during embryonal development of the human mid-brain have been ordered with the least mature at the foot and the most mature at the top. Strikingly, the wild-type LUHMES cells more closely correlated with the more immature neurons, whereas the H116 cells showed greater similarity to more mature cell types. Representative enrichment plots for the analysis are shown (Fig. S4C).

The subset of genes that contributed most to these distinctions, were identified by leading-edge analysis (LEA; Supplementary Data 5) with 493 genes contributing to H116 phenotype and 503 genes to WT phenotype. Using profiler, we associated those genes with KEGG pathways (Fig.S4D, Supplementary Data5). The WT phenotype was enriched in terms such as the cell cycle, DNA replication, and various cancers (presumably reflecting growth-related activities). In contrast, the H116 phenotype was associated with neuronal activity: synapses, calcium signaling, and hippocampal long-term potentiation. We note that several terms enriched for H116 correlate with PWS phenotypes; including hormonal regulation [(GnRH[42], parathyroid hormone[43], aldosterone[44]], insulin secretion[45], salivary secretion[46], circadian entrainment[29] and addictive behavior[47]. To test if the large number of genes deregulated in H116 can result from the modified activity of transcription factors (TFs) we performed g:Profiler multiquery analyses against TFs from the TRANSFAC database[48]. H116-linked genes were associated with multiple TFs including CTCF, which has been linked to imprinting[49] (Fig. S4E).

We conclude that the data suggest the model that loss of SNORD116 enhances some aspects of the developmental timing of neuronal cells.

## SNORD deletion does not clearly alter pre-mRNA splicing

Previous reports proposed roles for SNORD115, SNORD116 snoRNAs and extended snoRNA-related ncRNAs in alternative pre-mRNA splicing, acting directly via base-pairing with the target pre-mRNA or through protein sequestration[4,11,22,25,50]. We, therefore, analyzed our RNA-seq data for changes in pre-mRNA splicing using DEXSeq[51]. During differentiation in wildtype cells (comparing D00 with D15), we identified many changes in alternative splicing (for an example, see Fig. S8A). In contrast, comparing mutants with the wildtype at D15 identified a few candidate alternative splicing events (95 for H116 and 73 for H115). Inspection of the RNA-seq data in the UCSC genome browser revealed only 9 genes with clearly altered expression of a subset of exons, all with modest effects. In each case, differential expression was apparently not due to alternative splicing, but reflected the use of alternative transcription start sites (OLFM1, MYO15A, NRXN1, IQSEC1, NAV1, and GSE1) or alternative termination sites (LAMP2, GNAO1 and HERC2P3 pseudogene–a frequent breakpoint in PWS patients) (Fig. S8B). We also visually inspected the RNA-seq data for multiple other genes previously predicted or reported to be regulated by SNORD116: no

changes were confirmed in our data. The HTR2C gene[25] was not detectably expressed in LUHMES cells. We conclude that alternative splicing is unlikely to underlie the changes in gene expression in the SNORD115 or SNORD116 deletion cell lines.

## Changes in translation in SNORD deletion lines

SNORD115 and SNORD116-derived ncRNAs might also influence mRNA translation; potentially via post-transcriptional effects on mRNP composition and/or nuclear cytoplasmic transport. To assess this, the total proteome was determined for the wildtype, H115 and H116 cells at D00, D06, D10 and D15 using HPLC-coupled, tandem mass-spectrometry with data-independent acquisition (DIA) (Supplementary Data 1 and 6).

The wildtype proteomic data broadly replicated the results from transcriptomic analyses: (1) Large and rapid changes during the first interval of differentiation (Fig. S5A, B). (2) High similarity between undifferentiated wildtype and mutant cells, with progressive divergence during the differentiation process, including the advanced differentiation of mutant cell lines in PCA analysis (Fig. S5C, E). (3) Increased numbers of differentially expressed proteins (DEPs) between mutant and WT cells over the time course of differentiation (Fig. S5C). (4) Substantial overlap between proteins with altered abundance in H115 and H116 cells (Fig. S5D). Notably, in hierarchical clustering, the corresponding proteomic intensities and transcriptomic read counts were grouped together, indicating high similarity between transcriptome and proteome and supporting data accuracy (Fig. S5E).

Proteomic and transcriptomic data were compared to identify proteins showing increased or decreased abundance relative to the corresponding mRNA in mutant cells. As expected, the relationship between protein steady-state level and RNA steady-state level (P/R ratio) is highly variable between genes: The highest P/R ratios were for $TUBB4A$ ($3 \times 10^9$), $ACTA1$ ($9 \times 10^8$) and $MAP1LC3A$ ($6 \times 10^8$), with the lowest for $MT\text{-}CO1$ and $MT\text{-}CO3$ subunits (56 and 93, respectively) (Fig. S5G). Spearman's correlation between protein and mRNA expression levels in steady-state populations of undifferentiated cells $\rho = 0.55$ and in differentiating cells $\rho < 0.4$ (Fig. S5F).

For most individual genes P/R ratios were notably consistent between mutant and wildtype cells (Fig. S5H, I). However, 90 genes passed thresholds for significant changes in P/R ratios upon SNORD115/SNORD116 cluster deletions (see Supplementary Materials); 37 with increased P/R ratio and 53 decreased. These are presented as a heatmap in Fig. S6A, with selected genes shown in Fig. S6B. Several genes showed stable mRNA levels but pronounced differences in protein abundance.

We conclude that deletions of SNORD115 or SNORD116 can alter protein abundance relative to mRNA levels, with considerable overlap in targets. We speculate that direct or indirect, differences in mRNA packaging or export cause altered translation efficiencies.

## H116 vs H115 comparison pinpoints genes potentially crucial for PWS phenotypes

Our analyses reveal considerable overlap between transcripts with altered expression in the H115 and H116 cell lines (Fig. 5E). However, deletion of the SNORD116 cluster results in PWS, which is not the case for SNORD115[7]. We, therefore, focused on the subset of mRNAs and ncRNAs that might explain the specific association of the SNORD116 cluster with the PWS phenotype.

We focused on genes with significantly different expression in H116 vs H115, and H116 vs WT – altogether 44 genes. Upon manual curation, we further excluded those showing expression differences only in undifferentiated cells, with low expression or unconvincing genome browser traces (analysis details in Supplementary Materials) (Fig. 6A). We obtained a short list of 24 high-confidence genes, 4 ncRNAs, and 20 mRNAs; 15 were overexpressed and 9 underexpressed relative to WT cells. Since these RNAs are potential direct targets for SNORD116 snoRNAs, we searched for sequence complementarity

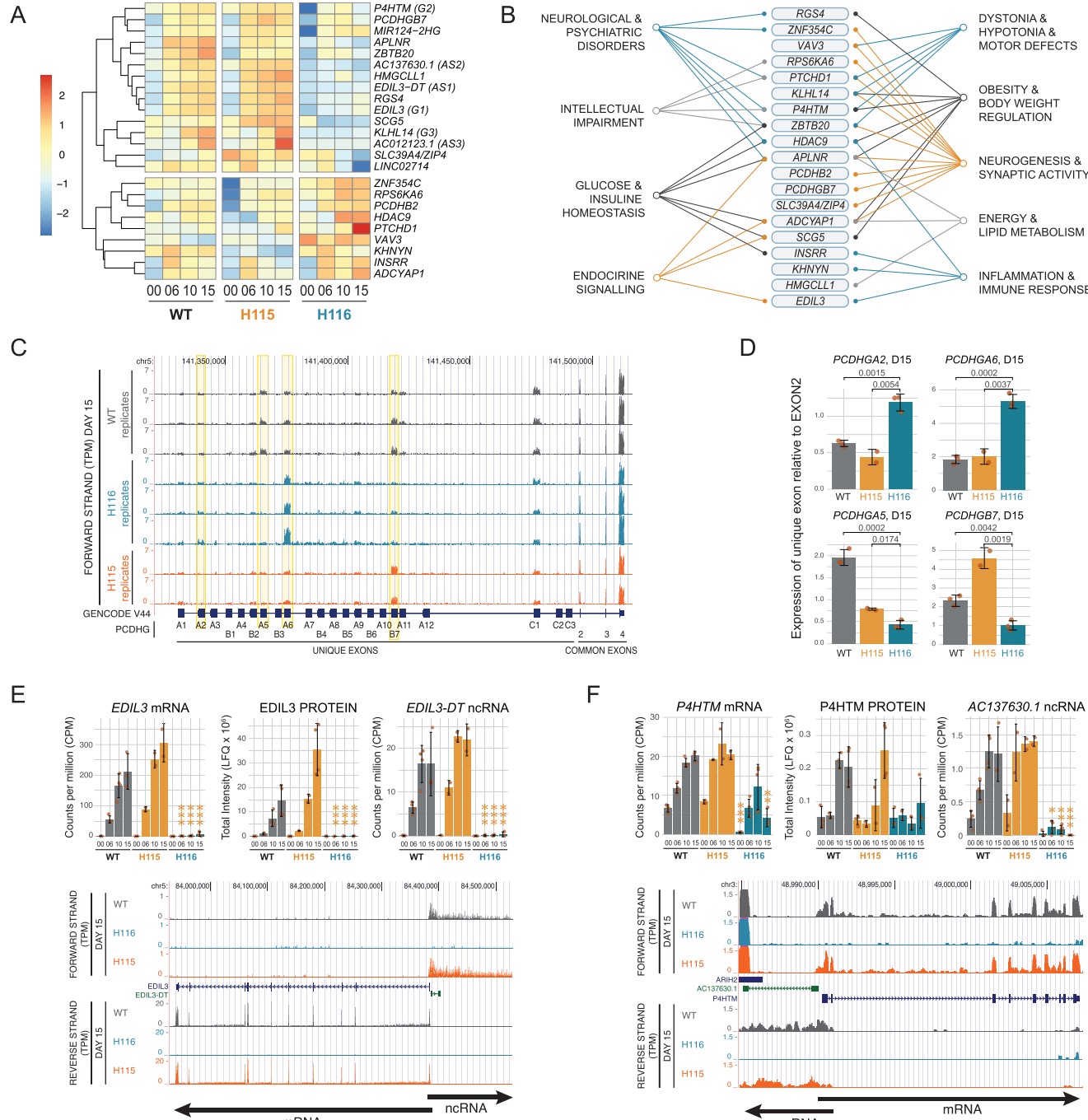

**Fig. 6 | SNORD116-specific genes may be crucial for PWS phenotypes. A** A heatmap representing the expression of SNORD116-specific genes in wildtype and mutant cells across differentiation. **B** Links between SNORD116-specific genes and PWS phenotype based on the published literature, supporting references are in the Supplementary Data 8. **C** UCSC genome browser view of transcription across clustered protocadherin genes gamma in wild-type and mutant cell lines at D15. Yellow boxes mark genes with the differential expression in H116 vs both wildtype and H115 and quantified in **D**. **D** Mean expression of unique exons of protocadherin gamma genes in RNA-seq data relative to common exon 2 at D15, marked yellow on **C**. Error bars represent standard deviation, *p* values obtained from Student *t* test.

**E, F** Mean expression of *EDIL3* (E) and *P4HTM* (F) mRNA, protein, and antisense non-coding transcript across differentiation. For transcriptomic data, statistical significance originates from EdgeR analysis (quasi-likelihood F-test followed by Benjamini-Hochberg correction for multitesting); for proteomic data, from DEP (empirical Bayes statistics followed by Benjamini-Hochberg correction for multitesting). Error bars represent standard deviation (SD). Number of biological replicates for each mutant and timepoint is provided in Supplementary Data 1. Source data, together with exact *p* values, are provided as a Source Data file. Below is the UCSC genome browser view of overlapping transcription from mRNA and ncRNA.

following the canonical rules for snoRNA-directed methylation, using PLEXY[52]. However, several hundred interactions that could potentially direct methylation were identified (Supplementary Data 7 lists the 631 most stable predicted interactions), making experimental validation challenging.

Many of the identified protein-coding genes are poorly annotated, with very limited functional information. However, they share some properties: Receptors or receptor regulators ADCYAP1, APLNR, RGS4, PTCHD1, INSRR, and P4HTM; Confirmed or predicted membrane proteins APLNR, EDIL3, INSRR, PCDHB2, PCDHGB7, PTCHD1 and

SLC39A4; Transcriptional regulators ZBTB20, ZNF354C, HDAC9, ADCYAP1 and VAV3. The identified genes also include two members of the protocadherin (PCDH) family. These cell-adhesion proteins play key roles in intercellular signaling and are crucial for self-avoidance and discrimination of self/non-self during brain development[53]. PCDH genes are expressed in 3 clusters, plus 15 non-clustered genes. No clear changes were seen for non-clustered PCDH genes, whereas PCDH alpha, beta, and gamma clusters showed altered expression patterns in H116 relative to the WT or H115 cells (Fig. 6C, D, and S8A and S8B). Visual inspection of genome browser data suggests that other clustered PCDH genes are affected by the SNORD116/115 deletions, but do not pass the strict differential expression thresholds in our bioinformatic analyses. Moreover, the literature review revealed a striking association of identified genes with PWS phenotype, e.g., glucose homeostasis, endocrine signaling, or intellectual impairment (Fig. 6B, for supporting references, see Supplementary Data 8).

Notably, 3 of 4 altered ncRNAs are transcribed from the opposite strand of mRNAs that also exhibited differential expression; EDIL3, P4HTM (Fig. 6E, F), and KLHL14 (Fig. S8C). In each case, the 5'-ends of the antisense (as) ncRNAs and mRNAs overlap. This organization might be expected to cause mutually exclusive transcription due to promoter occlusion, but both mRNAs and ncRNAs are lost in the absence of SNORD116. The fourth ncRNA (miRNA-2HG) is a host gene for miRNA-124, which is highly expressed in the brain and associated with nervous system development[54].

We conclude that the small number of specific changes in RNA levels in cells lacking SNORD116 pinpoint potential primary targets.

## Discussion

### Changes in SNGH14-derived ncRNAs during differentiation

Histone modification data indicates that the entire ~600 Kb, PWS-associated, SNHG14 region forms a single transcription unit. We followed the pattern of expression from the SNHG14 locus during neuronal differentiation (Fig. 2) and observed different transcript accumulation across different sections of the gene: (1) SNURF-SNRPN region increased expression upon differentiation, (2) spliced SNHG14 exons over the SNORD116 region showed little alteration and were always much more abundant than introns. This indicated that transcription and splicing of the SNHG14 primary transcript were essentially unaltered. Expression patterns for sno-lncRNAs, long ncRNAs containing two SNORD116 snoRNAs joined by a linker[10,11,31] were consistent with competing maturation pathways for SNORD116 snoRNAs and sno-lncRNAs. However, mature SNORD116 snoRNA expression was much higher than any other SNORD116-containing ncRNAs, suggesting that the mature snoRNAs may represent the major functional transcripts. (3) A quite different pattern was seen for the SNORD115 and UBE3-ATS regions, located further 3'. It seems likely that in undifferentiated cells, transcription terminates immediately prior to the SNORD115 region, as previously suggested[13]. During differentiation, termination readthrough increases, with SNORD115 generated following splicing.

### Altered gene expression in cells lacking SNORD116 or SNORD115

To better understand PWS-related changes, we created heterozygous deletion cell lines lacking expression from either the SNORD116 or SNORD115 cluster. Undifferentiated LUHMES cells express almost no SNORD115 and low levels of SNORD116, suggesting that they are unlikely to exhibit strong compensatory mechanisms following cluster deletion.

Deletion of the SNORD116 or SNORD115 clusters was not associated with altered accumulation of flanking exons within SNHG14, or of flanking genes within the PWS locus. There were, however, changes in the wider PWS region. Notably, mRNA from MAGEL2, located ~1.5 Mb upstream from SNORD116, was clearly under-accumulated. Mutations in MAGEL2 are causal in Schaaf-Yang syndrome, which shows striking similarities to the PWS phenotype[9,55]. We predict that reduced MAGEL2 mRNA and protein contribute to PWS phenotypes in individuals with SNORD116 microdeletions. However, we did not detect MAGEL2 protein in our proteomic data, so its depletion is unlikely to cause any of the observed defects.

Throughout the transcriptome, numerous genes and ncRNAs showed altered abundance in the mutant cell lines. In the wild type, changes in mRNA abundance during differentiation predominantly occur during the initial period (to D06) as the cells exit mitosis. In contrast, differences between the wildtype and mutants predominantly occurred at later stages, consistent with the time course of snoRNA accumulation. Differentially expressed genes were associated with molecular functions closely related to neuronal differentiation and functioning, such as axon guidance, regulation of membrane potential, or response to cAMP. Interestingly, they were also highly associated with the cell surface and extracellular matrix (GO term: cellular component), and 140 of 350 mRNAs downregulated at D15 in H116 cells are annotated as glycoproteins (https://www.uniprot.org/keywords/KW-0325). A recent report described the detection of glycosylated RNAs, including SNORD116[56]. The significance of this observation remains unclear, but it suggests a possible link between SNORD116 and glycosylation pathways and, potentially, cell-cell communication more generally. Those terms agree with the previous studies by Burnett et al.[57]. (e.g., glycoprotein, receptor, topological domain: extracellular) and Bochukova et al.[4] (e.g., axon development, calcium and chloride transport, and calcium signaling).

Principal component analysis of the RNA and protein expression data suggested that the mutant and wild-type cell lines might be on the same trajectory, but with the mutants advancing more rapidly. Comparison of mRNA levels in the wild-type and H116 cell lines to single-cell sequencing data of the differentiating human mid-brain supported this interpretation; cells lacking SNORD116 showed greater similarity to more mature neurons, while the wild-type resembled more immature cell types.

Effects seen on the loss of SNORD115 and SNORD116 could potentially be mediated by multiple mechanisms. Early effects of the deletions were observed in undifferentiated cells, even for H115 before the SNORD115 locus was expressed. These, and common H116 and H115 phenotypes, might reflect altered chromatin organization in and around SNHG14. Published chromatin conformation capture data obtained with Micro-C XL on the embryonic H1-hESC cells (Fig. S9) would be consistent with the region shown in Fig. 1 forming a topologically associated domain (TAD)[58,59]. The chromatin binding protein CTCF was identified in our LEA analysis of differential gene expression between WT cells and the H116 line, which appeared to have a more mature phenotype during neurodevelopment. Notably, CTCF regulates global transcription and 3D chromatin architecture, including acting as an insulator at domain boundaries[16]. It could be envisaged that the formation of a boundary element in SNHG14 and regulated transcription into SNORD115 might be controlled by CTCF. Later-acting mechanisms might involve mass-action effects of the snoRNAs as they accumulate to high levels, e.g., in RNA-protein condensates. We also cannot exclude the possibility that some phenotypic differences might result from comparing nucleofected and clonally selected mutant cells with the bulk population of non-treated wild-type cells.

Mature SNORD115 and SNORD116, as well as extended snoRNA transcripts, were previously reported or predicted to affect alternative pre-mRNA splicing[11,22,25]. We were, however, unable to reproduce any previously described splicing changes in our system by RNA sequencing performed at considerable depth. Few cases of differential intron accumulation were identified, with the clearest changes apparently reflecting altered transcription initiation or termination events. In marked contrast, many examples of apparent increased or decreased translation efficiency were discovered by comparison of transcriptome and proteome data. We speculate[9] that translation efficiency is altered

by changes in mRNP composition and/or nuclear/cytoplasmic transport following ncRNA loss.

## SNORD116-specific genes

Deletion of *SNORD115* or *116* clusters each caused multiple changes in the transcriptome of differentiating neurons, with considerable overlap. However, micro-deletion of *SNORD116* but not *SNORD115* has been reported to result in PWS. To identify the strongest links with the syndrome, we therefore examined changes specific to *SNORD116*, identifying a short list of mRNAs and ncRNAs. Most are poorly characterized, but the available data reveal neuronal-enriched expression and strong links to PWS phenotypes (see Supplementary Data 7 for a list of genes and selected references).

Of note is *ADCYAP1*, encoding the neuropeptide, pituitary adenylate cyclase activating polypeptide (PACAP), which is already markedly elevated after 6 days in H116 cells. ADCYAP1 has multiple signaling functions [reviewed in ref. 60] and was proposed as a potential master regulator for genes downregulated in the PWS hypothalamus[4]. We note that 6 of the 20 SNORD116-specific mRNAs (*ADCYAP1, KLHL14, HMGCLL1, PCDHGB7, ZBTB20, APLNR*) were differentially expressed in samples from PWS patients (Bochukova et al. 2018) but not always in the same direction. Moreover, five mRNAs were also affected by CTCF depletion during neuronal differentiation[61]: *ADCYAP1, EDIL3, MIR124-2HG, RGS4, SCG5,* and the clustered *PCDH* genes.

A small number of non-coding RNAs showed strongly reduced expression in H116. Strikingly, 3 out of 4 were transcribed antisense to the promoter regions of mRNAs that were also depleted: *EDIL3/EDIL3-DT, P4HTM/AC137630.1* and *KLHL14/AC012123.1*. In each case downregulation is detectable early, even in undifferentiated cells. The mechanism of coregulation remains unclear, but we speculate that the ncRNAs play a role in mRNA regulation, conceivably by altering the chromatin structure and/or through R-loop formation[62].

In conclusion, these analyses reveal the time course of changes in RNA metabolism during differentiation in a tractable neuronal cell line. We identify a modest number of candidate genes as potentially directly altered by loss of *SNORD116* in a disease model system.

## Methods

### Cell culture

LUHMES cells (ATCC cat# CRL-2927) (kindly supplied by A. Bird) were cultured according to published protocols[38,63]. Briefly, cells were grown on poly-L-ornithine (PLO), and fibronectin precoated dishes, and for long-term differentiation, cells were seeded on the Nunclon Delta-treated plates (ThermoFisher Scientific); for proliferation in Advanced DMEM/F12 (ThermoFisher Scientific, 12634028) with the addition of L-Glutamine, N-2 supplement (ThermoFisher Scientific, 17502048) and βFGF (R&D Systems, 4114-TC-01 M); for differentiation in Advanced DMEM/F12 with the addition of L-Glutamine, N-2 supplement, GDNF (R&D Systems, 212-GD-050), cAMP (Sigma-Aldrich, D0627-1 G) and doxycycline. Cells were differentiated in two steps: For pre-differentiation, cycling (D00) cells were seeded at $2.5 \times 10^6$ cells per T75 dish and grown for one day in a proliferation medium. This was exchanged for a differentiation medium and the cells were grown for two more days. On day 2 of differentiation, cells were trypsinized, counted, and seeded at $6 \times 10^6$ per 10 cm dish, starting the second step of differentiation. This is important because the differentiating neurons are very sensitive to cell density. During the subsequent differentiation process, half of the medium was changed every second day. Cells were taken for analysis 6 (D06), 10 (D10), and 15 (D15) days after the initial exchange to the differentiation medium. After D15 the LUHMES cells became sensitive to detaching from the dish, in which case they were discarded.

### CRISPR

To delete *SNORD115* and *SNORD116* clusters, Alt-R CRISPR-Cas9 system from IDT was used: two crRNA:tracrRNA guides (for sequences see Supplementary Data 7) for upstream and downstream cleavage complexed with Alt-R Cas9 nuclease 3NLS (ThermoFisher Scientific, 1074182) were prepared using following protocol: 0.5 μl crRNA-U [200 μM], 0.5 μl crRNA-D [200 μM] and 1 μl tracrRNA-ATTO [200 μM] were annealed in the PCR machine: 5 min at 95 °C, ramp −0.1 °C/sec to 25 °C. To deliver preassembled complexes and ssODN as a repair template, LUHMES cells were transfected by Nucleofection in a Nucleofector II device (Lonza) using a Basic Nucleofector Kit for primary neurons (Lonza, VAPI-1003) as described[63]. For each nucleofection reaction, the following proportions were used: $2 \times 10^6$ cells, 100 μl nucleofector solution and 5μl of mix: 1.2 μl (120 pmol) crRNA:tracrRNA, 1.7 μl Cas9 (104 pmol), 0.5 μl ssODN [100 μM] and 1.6 μl PBS. 48 hours post nucleofection cells were FACS sorted into 96-well plate for the isolation of clones. We isolated 54 clones (29% survival rate) with potential *SNORD115* deletion and 48 clones (25% survival rate) with *SNORD116*. Isolated clones were tested by PCR and by northern blot hybridization against SNORD115 and SNORD116 after neuronal differentiation. From CRISPR-SNORD115 we obtained two homozygotes and 3 heterozygotes, only one of them non-expressing SNORD115, i.e., with the deletion in paternal chromosome (H115-2/26), which was used for RNA- seq experiments. As the *SNHG14* gene is never expressed from the maternal chromosome, we initially included one homozygous mutant (D115) in the analyses. However, this cell line subsequently gave quite different results from all the heterozygotes at the level of transcriptome (in many cases appearing to be more similar to the wild-type). We therefore excluded it from further experiments and analyses used for final publication, other than the analyses of rRNA maturation. From CRISPR-SNORD116 we did not obtain any homozygotes and 3 heterozygotes, 2 of which were tested and shown not to express SNORD116; i.e., carrying deletion in paternal chromosome (H116-1/1, H116-2/15).

### Northern blot

Depending on the size of the RNA of interest we used two different kinds of protocols. For snoRNAs, 7SL and U1:10 μg total RNA was denatured in formamide loading dye and resolved on the 6% TBE-Urea gel (Novex, ThermoFisher Scientific) in 1× TBE buffer, until the Bromophenol Blue dye left the gel. To verify the even loading of the samples, the gel was stained with SYBRSafe (ThermoFisher Scientific) and scanned in the FLA-5100 scanner (FujiFilm). RNA was transferred to the Nylon Hybond-N+ membrane (RPN303B; GE Healthcare) by wet electro-transfer using BioRad MiniProtean System, for 1 hour at 30 V. After the transfer, RNA was crosslinked to the membrane with UVC in Stratalinker. Prehybridization was done in UltraHyb-Oligo (ThermoFisher Scientific, AM8663) for 2 hours at 42 °C. Probes were hybridized overnight (5 pmol) in 15 ml UltraHyb-Oligo at 37 °C. After washing, the membrane was exposed to the storage phosphoscreen (BAS-MP2040, Fuji). After overnight exposure, the screen was scanned in an FLA-5100 scanner.

For rRNA and lncRNAs: 2 μg total RNA was resolved on 1% agarose gel with TRI/TRI buffer and overnight capillary transfer onto BrightStar-Plus Positively Charged Nylon Membrane (ThermoFisher Scientific, AM10104). The detailed method and hybridization conditions are published[64]. rRNA probe sequences were taken from ref.[65,66]. A full list of probes can be found in Supplementary Data 7.

### RNA-seq libraries

In the RNA-seq analysis, we included 2 heterozygous H116 clones, 1 homozygous D115 clone, 1 heterozygous H115 clone, and a bulk population of wildtype LUHMES cells (not nucleofected or clonally selected). All the types of cells were grown on 10 cm dishes and lysed in 6 ml TRIZOL (15596026), frozen in two 3 ml aliquots. After phase separation, total RNA was collected in the aqueous phase, mixed with 2 volumes of 100% ethanol, and further purified with Zymo Direct-Zol MiniPrep Plus kit (Zymo Research, R2072). 6 μg total RNA was treated

with DNase RQ1 (Promega), purified with RNA Clean & Concentrator-5 kit (Zymo Research, R1013) and tested for integrity on Bioanalyzer using RNA 6000 Nano kit (Agilent, 5067-1511). Ribosomal RNA was depleted from 0.8 μg total RNA using NEBNext rRNA Depletion Kit (New England Biolabs, E6350L) following the manufacturer's protocol. RNA-seq libraries were prepared with NEBNext Ultra II Directional RNA Library Prep Kit for Illumina (New England Biolabs, E7765), and their quality was assessed on Bioanalyzer using High Sensitivity DNA assay (Agilent, 5067-4626). RNA-seq libraries were sequenced by BGI Genomics. Each sample represents a biological replicate, with no technical replicates included. A list of samples and number of sequencings reads for each of them are provided in the Supplementary Data 1. This list includes also samples prepared from homozygous mutant D115. Although, surprisingly, this cell line is quite different from the remaining heterozygous cell lines, the observed differences may be meaningful and will potentially aid understanding of the role of the PWS cluster in the regulation of gene expression. We are, therefore, making the full datasets of samples publicly available.

## RNA-seq analysis

Sequencing reads preprocessed with **flexbar** (adapter trimming and quality filtering)[67] were aligned to the genome (GRCh38 downloaded from Ensembl) with **STAR**[68] (version=2.7.3a, --outMultimapperOrder Random) and aligned to the genomic features using **featureCounts** (version: 2.0.0, parameters: -p -t exon -g gene_id -Q 10 -s 2)[68] and annotation from GENCODE (gencode.v34.annotation.gtf; evidence-based annotation of the human genome (GRCh38), version 34 (Ensembl 100) from 2020-03-24[69];). All samples show high Spearman correlations (Supplementary Data 1) calculated with corrplot package[70] in RStudio (R Core Team; 2021; R: A language and environment for statistical computing. R Foundation for Statistical Computing, Vienna, Austria. https://www.R-project.org/). Differential expression analysis was performed using **EdgeR** package[71] in RStudio. All the samples were combined in one DGEList, filtered by expression and normalized together, data dispersion was estimated with experimental design (-batch + group) with group representing type of mutation at each timepoint e.g. WTD00, H115D15. Testing for differential expression was performed pairwise – mutation type vs WT strain for each differentiation time point with glmTreat function with the threshold of lfc=log2(1.5), and H116 vs H115 with the threshold lfc=log2(1.2) for increased sensitivity (typical glmTreat thresholds are between 1.1 −1.5). A list of differentially expressed genes is available in Supplementary Data 3. The above analysis was confirmed using alternative strategy: VOOM followed by limma[72] either eBayes function with significance threshold of lfc = log2(2) or more stringent treat function with lfc = log2(1.5). The outcome of this comparison in summarized in Supplementary Data 1.

Functional enrichment analysis for differentially expressed genes (DEPs) between H116 and wild-type cells from all the differentiation stages combined was performed with g::Profiler (Raudvere et al. 2019) with the following parameters: data source: GO ontology: BP, MF CC; Statistical domain scope: all the genes included in EdgeR analysis, custom over all known genes; Significance threshold: g_SCS; user threshold: 0.05; electronic annotations [IEA] included. Most meaningful terms, selected manually with support from Revigo tool[73] are included in the plot created in RStudio with ggplot2 (cite: Wickham H (2016). ggplot2: Elegant Graphics for Data Analysis. Springer-Verlag New York. ISBN 978-3-319-24277-4, https://ggplot2.tidyverse.org).

PCA analysis was performed with prcomp function in RStudio using scaled log2(CPM) values and visualized with autoplot[74].

Analysis of differentiation was performed similarly to what is described above, with the following differences: only samples from wild-type cells were included in the analysis, data dispersion was estimated with experimental design (-0 + diffStage). Testing for differential expression was performed pairwise between the consecutive

timepoints with glmQLFTest function with high threshold of lfc = log2(2).

## Gene clustering

Clustering of genes based on the expression profile during differentiation was performed on the filtered by expression and normalized data from the EdgeR analysis of wild-type samples described above. Average expression for each gene (CPM) for a given timepoint was calculated and normalized to the maximum expression for this gene, resulting in all the expression values falling in the range between 0 and 1. Those data were used as an input for k-means clustering with $k = 6$ giving the best resolution without creating too much redundancy in the expression profiles. Functional enrichment analysis for genes belonging to each cluster was performed with g::Profiler tool[75], online version, with the following parameters: multiquery of 6 sets of genes CL0 to CL5; data source: GO ontology: Statistical domain scope: all the genes from CL0-CL5 clusters, custom over annotated genes; Significance threshold: g_SCS; user threshold: 0.05; electronic annotations [IEA] not included. List of the genes belonging in each cluster, excluding those filtered out by g::Profiler algorithm is provided in Supplementary Data 2, together with the detailed outcome of the analysis. Most meaningful terms, selected manually with some support from Revigo tool[73] are included in the plot created in RStudio with ggplot2[76] (https://ggplot2.tidyverse.org).

## Alternative splicing analysis: DEXseq

STAR mapped RNA-seq data were aligned with **featureCounts** (Version 2.0.3) against flattened GTF file, i.e. genes with overlapping coordinates are combined into one composite gene (e.g., gene_id "ENSG00000243485.5 + ENSG00000284332.1"). Flattened file was produced by the dexseq_prepare_annotation2.py function downloaded from Github (Vivek Bharwaj, Subread_to_DEXSeq, Oct 27 2018 https://github.com/vivekbhr/Subread_to_DEXSeq) as recommended by DEXseq manual, with gencode.v34.annotation.gtf from GENCODE as an input file. As featureCounts doesn't accept gene description longer than 256 bytes, 4 composite genes were removed from the analysis, among them *SNHG14* gene. featureCounts calculated reads mapping to exons and was run with the following parameters: -p –countReadPairs -f -O -Q 10 -s 2. Alternative splicing was analyzed with DEXSeq package[51] from Bioconductor project[77], independently for undifferentiated and d15 neurons, mutant vs WT cells, as well as for the differentiation of WT cells – WTD15_vs_WTD00, which was used as a positive control of the analysis.

## Calculating expression of *SNHG14* and PCDH exons

STAR mapped RNA-seq reads, were aligned with **featureCounts** against modified GTF file containing exclusively one type of features – exons, redundant exon annotations were removed (modified gencode.-v34.annotation.gtf from GENCODE). Exon coverage data (CPM) were normalized to the size of the library and filtered using EdgeR package. For *SNHG14*, average values for each exon for each differentiation stage were calculated and normalized to the exon length. Distribution of normalized average expression values for exons within *SNORD115* and *SNORD116* clusters was visualized with ggplot2. For PCDH genes from clusters alpha and gamma, we aimed to obtain relative expression of each gene within a cluster independent of total expression level. Therefore, we compared the expression of unique exon 1 for each gene normalized to the expression of common exon 2. As PCDH genes from cluster beta are monoexonic, expression of those genes comes from the original RNA-seq analysis. The statistical significance of the differences between samples was calculated using *t* test.

## GSEA/LEA analysis

Counts per million values (CPM) from EdgeR RNA-seq analysis (filtered and normalized to the size of the library) were used as input for the

GSEA[40]. Experimental data were tested for enrichment in gene sets from MSigDB[40], C8 collection: c8.all.v7.4.symbols.gmt using following parameters: permutation type: gene_set, number of permutations: 1000. For further analysis, only gene sets originating from mid-brain differentiation (MANNO_MIDBRAIN_PHENOTYPES)[41] were followed. Genes that contribute to the distinct phenotypes were identified by Leading Edge Analysis (LEA, utility from GSEA). Functional enrichment analysis was performed on those genes using g::Profiler tool[75], online version, with the following parameters: multiquery of two sets of genes WTd15-enriched and H116d15-enriched; data source: KEGG, TRANSFAC; Statistical domain scope: all the genes included in the GSEA analysis, custom over all known genes; Significance threshold: g_SCS; use threshold: 0.05; electronic annotations [IEA] not included. Detailed outcome of the analysis together with list of genes from LEA analysis, is provided as Supplementary Data 6.

## MS samples

All samples used for the MS analysis come from independent rounds of differentiation and are biological replicates. The exact number of replicates for each condition is stated in Supplementary Data 1. Samples were processed with modified FASP protocol[78]. Briefly, $0.5 \times 10^6$ LUHMES cells (number of cells calculated at d2 of differentiation) were lysed in 100 µl lysis buffer (25 mM Tris-HCl pH 7.5, 50 mM DTT, 0.1% Rapigest), incubated in a thermoblock for 5 min at 95 °C, mixing with 500 rpm, then allowed to cool to room temperature. Samples were sonicated at 4 °C in Bioraptor Pico (Diagenode) for 10 cycles, 30 sec on, 30 sec off. 50 µl of the sample was mixed with 200 µl buffer B (8 M urea, 100 mM Tris-HCl pH 7.5), transferred onto the Vivacon 500 30k spin columns (Sartorius), and centrifuged at $14,000 \times g$ for ~30 minutes until the buffer had all passed through. Proteins on the membrane were dissolved in 80 µl 100 mM iodoacetamide in 8 M urea incubated in darkness for 20 min and centrifuged until the buffer had gone through. Samples were washed twice with 80 µl 50 mM ammonium bicarbonate (buffer ABC) and centrifuged until dry. 100 µl Trypsin solution (10 µg/ml in ABC) was added and samples are incubated overnight at 37 °C. The next day, the peptide digest was collected by centrifugation into the collection tube. Membrane is rinsed with 80 µl buffer ABC and both fractions were combined. The peptide concentration was measured on Qubit with Qubit Protein Assay (ThermoFisher Scientific), and samples were acidified by adding 10ul of 10% TFA. C18-stage tips were prepared as described[79] and loaded with 10 µg tryptic peptides. StageTips, used to clean and concentrate the samples following digestion, were prepared as described[80]. Peptides were eluted in 40 µL of 80% acetonitrile in 0.1% TFA and concentrated down to 1 µL by vacuum centrifugation (Concentrator 5301, Eppendorf, UK). The peptide sample was then prepared for LC-MS/MS analysis by diluting it to 5 µL by 0.1% TFA.

LC-MS analyses were performed on an Orbitrap Fusion™ Lumos™ Mass Spectrometer (ThermoFisher Scientific, UK) coupled online, to an Ultimate 3000 HPLC (Dionex, ThermoFisher Scientific, UK). Peptides were separated on a 50 cm (2 µm particle size) EASY-Spray column (Thermo Scientific, UK), which was assembled on an EASY-Spray source (Thermo Scientific, UK) and operated constantly at 50 °C. Mobile phase A consisted of 0.1% formic acid in LC-MS grade water, and mobile phase B consisted of 80% acetonitrile and 0.1% formic acid. Peptides were loaded onto the column at a flow rate of 0.3 µL min⁻¹ and eluted at a flow rate of 0.25 µL min⁻¹ according to the following gradient: 2 to 40% mobile phase B in 180 min and then to 95% in 11 min. Mobile phase B was retained at 95% for 5 min and returned to 2% a minute later, until the end of the run (220 min).

Survey scans were recorded at 120,000 resolution (scan range 350–1100 m/z) with an ion target of 8.0e5, and injection time of 50 ms. MS2 DIA was performed in the orbitrap at 60,000 resolution, maximum injection time of 55 ms and AGC target of 1.0E6 ions. We used HCD fragmentation[81] with fixed collision energy of 30. From scan range 300–1000 m/z we used isolation windows of 17 m/z and default charge state of 3. The desired minimum for points across the peak was set to 6.

The DIA-NN software platform[82] version 1.8.1. was used to process the DIA raw files, and a search was conducted against the Uniprot database (released in July 2017). Precursor ion generation was based on the chosen protein database (automatically generated spectral library from the protein database used) with deep-learning-based spectra, retention time, and IMs prediction. Digestion mode was set to specific with trypsin allowing maximum of one missed cleavage. Carbamidomethylation of cysteine was set as fixed modification. Oxidation of methionine, and acetylation of the N-terminus were set as variable modifications. The parameters for peptide length range, precursor charge range, precursor m/z range, and fragment ion m/z range, as well as other software parameters, were used with their default values. The precursor FDR was set to 1%.

## MS data analysis

Differential expression of proteins was performed using DEP package[83]. Samples were filtered for proteins that are present in all replicates of at least one condition (filter_missval(SE, thr = 0)), VSN normalized and missing values were imputed with MinProb method ($q = 0.01$) as most of the proteins are expected to be missing not at random (MNAR). Proteins are tested for differential expression using test_diff function and manually defined contrasts between conditions, i.e., mutation type and stage of differentiation, e.g., H116d10_vs_WTd10. Obtained $p$ values were corrected for multiple testing with Benjamini–Hochberg procedure using R stats package and all the proteins with p.adj ≤ 0.05 are treated as differentially expressed. A list of differentially expressed and all quantified proteins is available in Supplementary Data 5.

For individual genes P/R ratios were surprisingly stable, especially when compared within the same stage of differentiation. For most genes, P/R fold difference between mutant and wildtype cells oscillated closely around 1 (Fig. S5H). Spearman's correlation between P/R ratios was also very high, $\rho > 0.9$ within the same stage of differentiation (Fig. S5I) and $\rho > 0.8$ between different stages (data not shown). We therefore used P/R ratio to identify potential instances of different post-transcriptional gene regulation in wildtype and mutant cells. For higher reliability, we focused on DEPs with the most statistically significant changes in protein expression and at least twofold difference in P/R ratio in at least two differentiation stages, for deletion of either SNORD cluster (Fig. S5I).

## Transcriptome-proteome analysis

Proteome (mean LFQ value for cell line at given differentiation stage, e.g., PROT_Mean_WTD00) and transcriptome data (mean CPM value for cell line at given differentiation stage, e.g. RNA_Mean_WTD00) after prior filtering and normalization steps described above, were combined in the same analysis.

Spearman correlations between steady-state protein and RNA expression were calculated and ranged from 0.55 for undifferentiated cells to about 0.32 at D06 when the cells were still dynamically adjusting to changes connected with differentiation. Correlation values were dependent on the differentiation stage and very similar for all the cell lines (Fig. S6E). Despite moderate correlation of RNA-seq and MS data, clustering of combined datasets associated PROT and RNA data originating from the same cell type and differentiation stage (Fig. S6F). Because of the difference between LFQ and CPM values, clustering was performed on the values scaled independently for PROT and RNA data (R, scale function). Clustering and plotting heatmap was performed with Pheatmap package (R, Version 1.0.12, Raivo Kolde, Pheatmap, 2019-01-04, https://CRAN.R-project.org/package=pheatmap). The ratio between protein and RNA expression (P/R) varies by six orders of magnitude from ~$10^2$ up to ~$10^8$ (for tubulin TUBA4A). To test if this wide range of values reflects real conditions in the cells or just high noise in our sequencing data we compared P/R ratios for all

genes, between cell lines at specific differentiation stages (P/R$_{MUT}$)/(P/R$_{WT}$) (Fig. S6). This value is quite stable as visible from a narrow distribution of values around 1 (full range of values: 0.003–81), suggesting that P/R ratio is a characteristic feature of a gene at a given differentiation stage and that can be utilized to test the hypothesis that ncRNAs from PWS locus influence post-transcriptional gene expression. We consider that direct influences of the ncRNAs on the stability of multiple proteins are unlikely.

We focused on genes that show changes in P/R ratio of at least two-fold upon deletion of the SNORD115 or SNORD116 cluster (Fig. S6I) for at least two differentiation stages and are identified as DEPs. This allowed us to limit the analysis to the most reliable subset of genes, with the protein expression stable enough to pass the statistical criteria of differential expression analysis. For all those genes, we created a heatmap using ComplexHeatmap package[84].

**RT-PCR analysis**

1 µg of DNase RQ1-treated total RNA samples prepared for RNA-seq analysis was reverse transcribed using SuperScript IV (ThermoFisher Scientific, 18090050) and random hexamers (Promega). PCR reaction was prepared with Premix EX Taq II mastermix (TaKaRa, RR82WR) with intercalator-based TB Green quantitation, and ROX Reference dye as a control; and run in Mx3005P machine (Stratagene). Expression of each gene was normalized to *ACTB*, that, based on the RNA-seq data, maintains stable expression during the differentiation process.

For relative D00 and D15 quantitation of SNORD115 and SNORD116 expression, we used three independent samples from WT cells, and a standard curve was prepared from one of the D15 samples. For the validation of RNA-seq data, we directly compared one sample used previously in our RNA-seq analysis with one set of samples from independent differentiation. Error bars represent standard deviation from technical replicates.

**Reporting summary**

Further information on research design is available in the Nature Portfolio Reporting Summary linked to this article.

## Data availability

The data supporting the findings of this study are available from the corresponding authors upon request. Information on the high-throughput data samples analyzed in this paper is provided in Supplementary Data 1. RNA-seq data have been deposited in the Gene Expression Omnibus (GEO) database under the accession number GSE277484. The mass-spectrometry proteomics data have been deposited to the ProteomeXchange Consortium via the PRIDE[85] partner repository with the dataset identifier PXD057208. Source data are provided with this paper.

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

## Acknowledgements

We thank Shaun Webb and Sang Seo for advice on bioinformatics, Tatsiana Auchynnikava for assistance with proteomics, Justyna Cholewa-Waclaw, Ruth Shah, and Adrian Bird for the LUHMES cell line, Emily Osterweil and Susana Ribeiro Dos Louros for consultations on the neuronal cell biology. We thank Dhanya Cheerambathur, Chris Sibley and colleagues for critical reading of the M.S. We thank Timofey Rozhdestvensky for the insightful comments on the preprint version of our manuscript on BioRxiv. D.T. was supported by a Wellcome Principal Research Fellowship (077248). A.H. was supported by the Foundation for Prader-Willi Research (FPWR), and T.W.T. was supported by the Polish Ministry of Science and Higher Education Mobility Plus program (1069/MOB/2013/0). This work was supported by funding for the Wellcome Discovery Research Platform for Hidden Cell Biology (226791). We gratefully acknowledge support from the Proteomics and Bioinformatics cores.

## Author contributions

A.H. and D.T. conceived the project and wrote the manuscript. A.H. and C.S. performed experiments. A.H., T.T., C.S., and D.T. analyzed data. All authors edited and reviewed the manuscript.

## Competing interests

The authors declare no competing interests.
