## [Transparent Peer Review file · Nature Communications]

Roles of SNORD115 and SNORD116 ncRNA clusters in neuronal differentiation

Corresponding Author: Professor David Tollervey

Version 0:

Reviewer comments:

Reviewer #1

(Remarks to the Author)

In this manuscript, Helwak et al focused on two human snoRNA gene clusters (SNORD115 and SNORD116) positioned at Chr15q11q13 and whose expression is regulated by genomic imprinting (their expression is restricted to the chromosome of paternal origin). Abnormalities in the functioning of paternal alleles at this epigenetically-regulated chromosomal domain are causally linked to a rare neurodevelopmental syndrome: the Prader-Willi syndrome. The studies of rare patients suggest that defects in snoRNA expression, and in particular that of SNORD116, may play a key role in the etiology of the disease. The mode of action of these two "orphan" snoRNAs remains however largely enigmatic. By combining genome editing using the CRISPR/Cas9 method and genome-wide transcriptomic and proteomic analyses, Helwak et al describe the molecular consequences of loss of SNORD116 or SNORD115 expression in the Lund human mesencephalic (LUHMES) cells. One of the major findings of their manuscript is that the absence of snoRNAs is associated with alterations in the timing of neuronal differentiation. More precisely, snoRNA-KO cells appear "more mature" than WT controls. The authors also mention the existence of FBBL1, an intronless gene related to Fibrillarin, but whose expression is mainly observed in the brain. Finally, the authors also provide experimental evidence suggesting that the biogenesis of SNORD116 and SNORD115 is regulated during neuronal differentiation, with stabilization of the mature forms of SNORD116 as well as transcriptional readthrough mechanisms for SNORD115.

Although this study does not enable us to understand the mode of action of SNORD115 and SNORD116, I find the subject interesting and relevant, with most of the conclusions in line with the experimental data generated, even if certain conclusions seem hasty to me and would have deserved more experimental development. Before considering publication of this study, however, I feel it is important for the authors to provide answers to my questions/comments, which are listed below and which I believe will improve the quality and readability of the manuscript. It also seems to me that the discussion could be reduced, and that the summary could also be made more attractive, notably for readers unfamiliar with this theme (optional).

Major comments

1- Characterization of human cell lines lacking SNORD115 and SNORD116 gene arrays.

"We precisely deleted regions of SNHG14 gene containing the snoRNA clusters from only the paternal (expressed) chromosome, using CRISPR in LUHMES cells (Fig. S2A)».

I imagine that the authors have not formally demonstrated this, but that they assume it because only one of the two alleles has been deleted, the LUHMES cells are diploid and SNORD115 and SNORD116 are no longer expressed. By the way, are we sure that the 15q11q13 locus is present in only two copies in LUHMES cells?

As I understand it, only 2 clones deleted for SNORD116 genes and a single SNORD115-deficient clone have been isolated. In this context, the authors should specify the total number of isolated clones that were screened.

"Intriguingly, some changes in mRNA accumulation were detected in undifferentiated cells lacking SNORD115 or SNORD116 clusters (106 in H116 and 38 in H115, 21 common). This was unexpected, especially for H115 cell line, as undifferentiated LUHMES cells apparently lack transcription across the SNORD115 cluster".

“We initially included one homozygous mutant (D115) in the analyses. However, this cell line subsequently gave quite different from all the heterozygotes at the level of transcriptome (in many cases appearing to be more similar to the wild-type). We therefore excluded it from further experiments and analyses used for final publication, other than the analyses of rRNA maturation”.

I really appreciate the fact that the authors raise explicitly these counter-intuitive observations. Given that the authors studied a limited number of clones, and in the absence of rescue experiments, it seems to me that we cannot rule out the possibility that some of the changes observed might be due to clonal effects (“compensatory pathways”) and/or off-target effects introduced by the CRISPR-Cas9 method (“undesired cleavages”). In this framework, what was the rationale behind the authors' decision to choose the heterozygous SNORD115 clone over the homozygous one?

Importantly, do WT cells correspond to cells that have (i) been transfected; (ii) been isolated by cell sorting and (iii) undergone clonal selection, but do not bear a deletion (at least on the paternal allele). Unless I am mistaken, this is not specified in the manuscript; and given the concerns raised above, I think it's very important to avoid the detection of changes that are not directly related to the targeted deletion.

2-Differentiating LUHMES cells express a homologue of core snoRNP protein Fibrillarin.

That FBLL1 gene is only expressed in brain is very intriguing and unexpected. However, I am not sure what point there is in mentioning it in this manuscript, since the authors indicate that they wish to develop this aspect in another study. In this context, the reader might think that SNORD115 and SNORD116 are preferentially associated with FBLL1 in brain, which to my knowledge has never been demonstrated. Moreover, a role in stabilizing SNORD116 seems unlikely since SNORD116 is expressed in all human tissues analyzed to date (n= 20 human tissues, see Vitali et al 2009 (PMID:20016068)). I propose that the authors withdraw this section, which, in the absence of strong experimental data, does not provide any relevant information as to the regulatory function of SNORD116 or SNORD115.

3- Transcriptome and Proteome Analyses

As I understand it, the authors did not attempt to further validate the mis-expression of some differentially-expressed genes by another technique (RT-qPCR) and using other RNA samples. It is a pity because it could have made the observations more robust and therefore more convincing.

Numerous publications have reported the transcriptional and/or post-transcriptional consequences of a gain- or loss-of-function of SNORD116 or SNORD115 genes, or even proposed direct RNA targets, whether in human cell models or in mouse tissues. Among these publications, we can mention Ding et al (2010) PMID: 20195375, Pace et al (2020) PMID: 32365348, Burnet et al (2017) PMID: 27941249, Bochukova et al (2018) PMID: 29590610, Hebras et al (2020) PMID: 33016258, Baldini et al (2022) PMID: 34893870, Falaleev et al (2015) PMID: 26220404; Powel et al (2013) PMID: 23771028. I think it would be interesting for the authors to compare their observations with those already available in the literature, perhaps in the form of a table or Venn diagram. Indeed, there is a lot of disparity in the literature, making it difficult to understand the precise role of SNORD115 and SNORD116. I am obviously aware that it is not always easy to compare different biological contexts but it is important so that the reader can form his own opinion of the state of the field and its relative complexity.

Finally, it is not clearly stated whether changes in gene expression, as judged by RNA-seq or proteomics, alter the morphology of differentiated cells and/or their behaviour (eg, neuronal activity). In my view, it would be informative to describe the cellular consequences, if any, of the molecular alterations described.

4- Regulation of SNORD116 and SNORD115 synthesis during neuronal differentiation.

Page 7 - “We therefore predict that the apparent extension of the SNHG14 transcripts into the region surrounding the SNORD115 cluster reflects regulated read-through of a termination site located 3' to SNORD116”.

The presence of a transcriptional stop signal positioned between SNORD116 and snoRD115 gene arrays was already predicted in Vitali et al 2009 (PMID: 20016068). That the spatiotemporal expression profiles of SNORD115 and SNORD116 do not perfectly overlap during *in vitro* neuronal differentiation or brain development was also described in Vitali et al 2009 (PMID: 20016068) and Landers et al 2004 (PMID: 15226413). In human, SNORD116 and IPW genes are ubiquitously-expressed while SNORD115 gene expression is brain-specific (neuronal expression), as shown in Vitali et al 2009 (PMID: 20016068). More importantly, it has been recently shown that a boundary element overlapping IPW region prevents, in a tissue-specific manner, the expression of SNORD115 and UBE3A-ATS in non-neuronal cells (Martins-Taylor et al (2014) PMID: 24363065; Hsiao et al 2019 (PMID: 30674673)). The above-mentioned references must be quoted and discussed accordingly.

Page 7 – ‘We conclude that the region encoding SNORD116 is well transcribed in undifferentiated cells. The accumulation of exon regions relative to introns indicates that splicing is functional, suggesting that the failure in mature SNORD116 accumulation reflects instability’.

I find this observation interesting but without any further experimental proofs, it seems to me highly speculative and not so convincing. Indeed, the authors demonstrate the expression of the SNORD116 gene array by sensitive method (RNA-seq) while the expression of the fully processed SNORD116 is monitored by Northern blot. In other words, failure to detect

SNORD116 in non-differentiated cells does not mean that it is not expressed. Moreover, we have no idea of the precursor-product relationship, then we can easily imagine a time lag between the detection of the primary transcription product and the nucleolar accumulation of SNORD116 RNP particles. In other words, there is no need to invoke post-transcriptional regulation. Do small RNA-seq data exist that would allow a more relevant and direct comparison? Can the authors detect SNORD116 containing sno-lncRNAs by Northern blot analyses?

Minor comments

Page 3 – “The smallest remove a ~70 Kb region of SNHG14 (snoRNA host gene 14), in which 29 tandem introns each encode the small nucleolar RNA (snoRNA) SNORD116 (Fig. 1)”.

In most microdeletions described so far, SNORD109A et the poorly characterized IPW ncRNA are also lacking.

Page 3 – “SNHG14 generates a very long non-protein coding RNA (lncRNA) with a predicted primary transcript around 600 Kb in length including 145 annotated introns”.

I recommend that the authors quote the original publication, namely Runte et al (2001) PMID: 11726556

Page 4 – “Box C/D snoRNAs generally form extended base-paired interactions that precisely target the nucleotide located 4 base-pairs from the box D motif in the snoRNA”.

This sentence is very ambiguous. Ribose methylation is targeted at the 2'-hydroxyl of the nucleotide base paired to the 5th, and not the fourth, nucleotide upstream of the D or D'.

Page 5 – “These could not be linked to direct snoRNA base-pairing”.

Have the authors systematically tried to search for potential base-pairings between SNORD115 and SNORD116 and transcripts whose abundance is altered in KO cells?

Page 6 – “We tested snoRNA expression over differentiation time course up to day 15, after which LUHMES cells become sensitive to detaching from the dish”.

What does it mean? Do cells die after 15 days of neuronal differentiation? I think it's important to clarify this sentence because most of the changes observed between KO and WT cells appear late in the course of neuronal differentiation.

Page 9 - “The maternal chromosome is transcribed on the opposite strand and was left intact”?

What does it mean “opposite strand”? I assume the authors are referring here to the expression of the UBE3A gene on the maternal allele, and not that of SNORD115 and SNORD116 clusters as a non-expert reader might understand. I think it's necessary to clarify this sentence to avoid any confusion.

Page 19 – “Moreover, mRNAs that are specifically affected show enrichment for neuronal function, including phenotypes related to features of PWS”.

I recommend that the authors mention a few examples.

Page 20 – “The effects on gene expression reported here are much more marked than in several previous analyses”.

To which studies do the authors refer? References must be added

Page 20 – “Previous mouse models for PWS have not fully recapitulated the human disease phenotype”.

To which studies do the authors refer? References must be added.

A targeted deletion of the SNORD116 cluster in SH-SY5Y neuroblastoma human cell line was already reported.

The authors must quote explicitly this work and discuss it accordingly.

In my view, the authors should quote some references in the last paragraph of the discussion section.

Although the information is available in the supplementary data, I think it is informative to indicate the number of biological replicates analyzed in the full manuscript.

Reviewer #2

(Remarks to the Author)

This was an interesting and potentially important study designed to investigate the impact s paternal loss of the SNORD116

and SNORD115 snoRNA clusters during differentiation of human LUHMES cells that can be differentiated into dopaminergic neurons. The authors describe a progressive snoRNA accumulation, describe a pseudogene transcript of fibularin (FBLL1) whose transcription is anti-correlated with the loss of FBL (involved in C/D snoRNA packaging) during neuronal differentiation. Using CRISPR/Cas9 generated heterozygous deletion models in LUHMES, they further describe changes to the transcriptome and proteome as a result of differentiation and genotypes. Overall, while there are some novel results and some convincing parts (like the deletion models), there is an overall lack of statistical rigor, lack of important controls, and the major conclusions are overstated.

Major concerns:

1. Carrying out differentiated LUHMES neuronal cultures beyond 1 week in culture raises concerns that some of the changes in transcriptome and proteome may be a result of diminished cell viability rather than differentiation. Measurements of live cell counts, reactive oxygen species (ROS) and/or apoptosis should be included for the experiments.
2. The DEG analysis was performed using EdgeR, which is based on the statistical assumption that most genes in the genomes are unchanged, which is likely not the case for either the differentiation time course or the effects of H116 and H115 mutations. These DEG analyses should be repeated with a more flexible model, as in limma+voom.
3. The k-means clustering approach is somewhat rudimentary way of clustering genes by coexpression. A Weighted Gene Co-expression Network Analysis (WGCNA) approach would be better for finding groups of genes that change over time and by genotype. (Langfelder, P. and Horvath, S., 2008).
4. The statement "Fig. S3). Comparison of D00 with D10 showed some reduction in pre-rRNA abundance during differentiation, consistent with the exit from cell division." Was not supported by any quantification or statistics. These results are not apparent by looking at the blots.
5. There are also no statistics to back up the conclusions being made about transcript and protein levels in Fig 5 C-E.
6. The results in Fig 6D-F, showing an increase in number of DEGs in H116 and H115 compared to WT at D10 and D15, could be the result of a reduction in cellular health and viability in these cultures rather than differences in differentiation, as the authors conclude. Cell stress in the cultures is implicated by the vast reduction in transcript levels compared to WT LUHMES at the same time points.
7. This unexpected result stated in the Results is also a concern: "Intriguingly, some changes in mRNA accumulation were detected in undifferentiated cells lacking SNORD115 or SNORD116 clusters (106 in H116 and 38 in H115, 21 common). This was unexpected, especially for H115 cell line, as undifferentiated LUHMES cells apparently lack transcription across the SNORD115 cluster." An alternative explanation for this phenomenon is that some of the DEGs are due to inter-individual differences in gene expression between different cell lines grown in separate flasks rather than SNORD-dependent. To address this, the authors should examine a comparison of separate WT LUHMES cell cultures compared to each other for a background level of DEGs seen independently of genotype differences. Since the genotype differences observed are quite subtle in this study, this WT-WT control is especially important to rule out possible background noise.
8. This major conclusion is overstated, based on the limited data shown in support and lack of cell viability controls: "We conclude that the data support a model that the loss of SNORD116 advances developmental timing."(Discussion) and "Most changes in mRNA and protein abundance appeared relatively late in development..."(Abstract) Comparisons to in vivo fetal brain transcriptome and transcription factor maps are only correlative and the authors have not ruled out alternative interpretations of the results.

Reviewer #3

(Remarks to the Author)

In "Roles of SNORD115 and SNORD116 ncRNA clusters in neuronal differentiation" Helwak and colleagues report a comprehensive survey of changes triggered by differentiation in a neuronal model. Extensive comparisons are made between wild-type and mutant cell lines generated with genome editing. Minimal alterations are detected at the level of splicing. Changes in gene expression, post-transcriptional regulation, and transcription are reported. It appears that differentiation is more rapid in the mutant cell lines based on inference from molecular measurements. There is overlap for some of the targets suggesting a functional interaction between the snoRNAs. Intriguingly, loss of the snoRNAs results in increased activity of several TFs. This work is comprehensive and technically sound. There are limitations regarding lack of mechanistic insights into precisely how the snoRNAs function. However, the manuscript is well written and clear. This work will likely be of broad interest to the field. I have several comments and questions rooted in genuine curiosity that I suspect will be shared by other readers. I hope that addressing these comments will result in an improved manuscript:

1. Given the similarities between H116 or H115, can the authors comment on what the consequence would be from a double mutant? Does this happen in any clinically relevant instances and is the outcome more deleterious than the SNORD116 deletion? Perhaps this would have a stronger phenotype that would be useful for future mechanistic investigations.
2. How do SNORD115/116 influence transcription factor activity? Do the authors think this is a direct or indirect effect? The authors rule out many of the obvious explanations that could explain this effect so what options remain? While clearly outside the scope of this manuscript, it would be of interest to identify the RNA/protein targets/partners of SNORD115/116. This work provides an essential resource for gauging the best timepoint at which to attempt this key experiment and to provide essential negative control cell lines.
3. How plastic are the effects of SNORD115/116 loss? Would expression late in development reverse the effects on transcription factor activity? Can the authors comment on the temporal dynamics of SNORD115/116 function? Is there any evidence that would enable discussion of this point as it has implications for potential salubrious use of this information for Prader-Willi syndrome.

4. The FBLL1 connection is intriguing. What are the physiologic consequences of FBLL1 elimination? Do these phenocopy loss of SNORD115/116? If this is the basis of a separate study, that's fine but I think it would be worth noting if there are existing data on this point.

Minor points:

5. Abstract – apparently is used in back to back sentences. Suggest more diverse word choice
6. Page 18 discussion – LUHMES might be too low high. Suggest correcting typo to low.
7. Fig 8A, esp. right panel: it is quite difficult to find some of the replicates on the proteome PCA plot due to the dots overlapping. The symbols might be made smaller so nearby points do not obscure one another.
8. Page 14: typo "...least mature AT the foot and the most mature at the top."

Version 1:

Reviewer comments:

Reviewer #1

(Remarks to the Author)

I am grateful to the authors for their efforts and additional work during the revision process. I am generally satisfied with their responses to my questions and comments. The modifications made to the revised version have improved its readability. If the editorial committee of Nature Communications decides to accept this manuscript for publication, it is essential to include critical elements in the discussion by addressing certain inherent limitations of this type of study.

In the full manuscript, the authors must mention and discuss that WT controls correspond to parental cells that did not undergo the same transfection and clonal selection process as the edited cells. Therefore, we cannot exclude the possibility that some differences between WT and mutated cells are related to other factors associated with the experimental procedure. Additionally, the initial version of the manuscript stated that KO115 homozygotes resembled WT more than heterozygotes. Unless I am mistaken, this counterintuitive information is no longer mentioned in the revised version. For transparency and to allow readers to form their own opinions, it is crucial that this information is mentioned in the full manuscript (Do some of the observed differences genuinely result from the absence of SNORD115 itself?). This could be achieved by adding a single sentence, without needing to show experimental data.

In my opinion, the part concerning FBLL1 does not integrate well with the rest of the study, especially since the authors do not provide, at the very least, an immunoprecipitation assay indicating that SNORD116 could associate with FBLL1. With the current data, the discussion of a putative role of FBLL1 in phase separation is intriguing but appears too speculative without experimental support. Furthermore, mentioning a link between FBLL1 and the stabilization of SNORD116 in the abstract risks leading to misinterpretation among readers. Nevertheless, I understand the authors' position and leave it to the editorial committee to make a decision on this matter.

Reviewer #2

(Remarks to the Author)

While I appreciate that the authors attempted to respond to all of the reviewers' concerns, the remaining problem is that very few of these suggestions for improvement were actually incorporated into the revised manuscript. As a result, the revised version of the manuscript is not an improvement. If anything, it is less impactful and more confusing because the authors have watered down the major conclusions in the abstract (such as the developmental timing one), yet they are still in the Results and Discussion sections. The suggestions for improving the rigor of the experiments through additional methods and approaches were not included in the manuscript. To be more specific, here are the remaining concerns:

Rev1 concerns:

The concerns regarding transcriptome effects that may have resulted from the lack of an appropriate control (mock transfected) and/or clonal and off-target effects were not addressed appropriately and my concern is about reproducibility. The reasons for wanting to keep the FBLL1 results in the manuscript were not appropriate. In the revised abstract, this section seems very disjointed from the rest of the manuscript.

The request for comparison of DEG results to other studies was not appropriate, as not all studies mention were hypothalamus (Powell et al is mouse cortex) and the overlap can be done at the level of GO terms and KEGG pathways which are less sensitive to bulk tissue concerns. If the authors are claiming that these cell line models are relevant to PWS, they need to provide evidence for that claim.

Figure R2 was not at all convincing in showing the SNHG14 transcript by Northern blot.

Rev 2 concerns:

The concern about cells at d10 and d15 should be able to provide RNA quality scores in the response.

The reason for suggesting additional methods for DEG analysis (Limma voom) and systems biology (WGCNA) was so that the authors could back up their major conclusion about developmental timing with additional methods. It appears that the authors were unable to back up the developmental timing claim with these methods because they were not included in the revised manuscript.

The concern about transcriptome results potentially being the result of cellular heterogeneity raised by both reviewers has not been appropriately addressed in the revised manuscript.

Rev3 concerns:

This reviewer had several suggestions and questions, none of which were incorporated into the revised manuscript.

Reviewer #3

(Remarks to the Author)

Thank you for addressing my comments.

We thank reviewers for their carefully considered comments. We have introduced numerous changes and have repeated many of the analyses where requested. In response to your comments we have expanded our analysis specifically to the SNORD116-specific genes. Due to the additional data and the space limitations, we had to remove some parts of the previous manuscript.

REVIEWER COMMENTS

Reviewer #1 (Remarks to the Author):

In this manuscript, Helwak et al focused on two human snoRNA gene clusters (SNORD115 and SNORD116) positioned at Chr15q11q13 and whose expression is regulated by genomic imprinting (their expression is restricted to the chromosome of paternal origin). Abnormalities in the functioning of paternal alleles at this epigenetically-regulated chromosomal domain are causally linked to a rare neurodevelopmental syndrome: the Prader-Willi syndrome. The studies of rare patients suggest that defects in snoRNA expression, and in particular that of SNORD116, may play a key role in the etiology of the disease. The mode of action of these two “orphan” snoRNAs remains however largely enigmatic. By combining genome editing using the CRISPR/Cas9 method and genome-wide transcriptomic and proteomic analyses, Helwak et al describe the molecular consequences of loss of SNORD116 or SNORD115 expression in the Lund human mesencephalic (LUHMES) cells. One of the major findings of their manuscript is that the absence of snoRNAs is associated with alterations in the timing of neuronal differentiation. More precisely, snoRNA-KO cells appear “more mature” than WT controls. The authors also mention the existence of FBBL1, an intronless gene related to Fibrillarin, but whose expression is mainly observed in the brain. Finally, the authors also provide experimental evidence suggesting that the biogenesis of SNORD116 and SNORD115 is regulated during neuronal differentiation, with stabilization of the mature forms of SNORD116 as well as transcriptional readthrough mechanisms for SNORD115. Although this study does not enable us to understand the mode of action of SNORD115 and SNORD116, I find the subject interesting and relevant, with most of the conclusions in line with the experimental data generated, even if certain conclusions seem hasty to me and would have deserved more experimental development. Before considering publication of this study, however, I feel it is important for the authors to provide answers to my questions/comments, which are listed below and which I believe will improve the quality and readability of the manuscript.

It also seems to me that the discussion could be reduced, and that the summary could also be made more attractive, notably for readers unfamiliar with this theme (optional).

We have altered the discussion and substantially changed the Abstract. In places we have reduced the length and/or transferred material to the SI, to better match the journal length requirements.

Major comments

1- Characterization of human cell lines lacking SNORD115 and SNORD116 gene arrays.

“We precisely deleted regions of SNHG14 gene containing the snoRNA clusters from only

the paternal (expressed) chromosome, using CRISPR in LUHMES cells (Fig. S2A)».

I imagine that the authors have not formally demonstrated this, but that they assume it because only one of the two alleles has been deleted, the LUHMES cells are diploid and SNORD115 and SNORD116 are no longer expressed. By the way, are we sure that the 15q11q13 locus is present in only two copies in LUHMES cells?

LUHMES cells are reported to be diploid (Shah *et al*, 2016), but normal karyotyping cannot exclude that PWS region is specifically duplicated. However, the RNAseq, PCR and Northern blotting data demonstrate that heterozygous deletion cell lines do not express ncRNAs from SNORD115 and SNORD116 clusters (paternal chromosome), whereas convergent non-overlapping UBE3A expression remains intact (maternal chromosome). We point this out explicitly in the revised MS (P9). If there is a duplicated cluster, it is either not expressed or was also deleted.

As I understand it, only 2 clones deleted for SNORD116 genes and a single SNORD115-deficient clone have been isolated. In this context, the authors should specify the total number of isolated clones that were screened.

That is true that number of obtained clones was low. Below are the details on the number of cells undergoing clonal selection.

Numbers:

2 million cells used per nucleofection, 2 x 96-well plates per deletion seeded by FACS 48 hours post nucleofection.

SNORD115:

- 54 clones (29%) survived sorting and got frozen, all tested by PCR.
- 2 homozygotes
- 3 heterozygotes with expected deletion size – 1 of them not expressing SNORD115 i.e. deletion from paternal chromosome

SNORD116:

- 48 clones (25%) survived sorting and got frozen, all tested by PCR.
- 0 homozygotes
- 3 heterozygotes with expected deletion size, 2 of them not expressing SNORD116 i.e. deletion from paternal chromosome, 1 clone was not tested.

We included this information in the Supplementary Methods, CRISPR paragraph.

“Intriguingly, some changes in mRNA accumulation were detected in undifferentiated cells lacking SNORD115 or SNORD116 clusters (106 in H116 and 38 in H115, 21 common). This was unexpected, especially for H115 cell line, as undifferentiated LUHMES cells apparently lack transcription across the SNORD115 cluster”.

The basis of this is unclear but could reflect the effects of deletions in the *SNHG14* gene itself or in long ncRNA transcripts. The region shown in Fig 1 is a single TAD (see Figure R1), so effects on the chromosome structure are also possible. CTCF is a candidate factor for mediating such effects. We briefly mention these points in the revised Discussion (P18).

Figure R1 common PWS breakpoints circled yellow

“We initially included one homozygous mutant (D115) in the analyses. However, this cell line subsequently gave quite different from all the heterozygotes at the level of transcriptome (in many cases appearing to be more similar to the wild-type). We therefore excluded it from further experiments and analyses used for final publication, other than the analyses of rRNA maturation”.

I really appreciate the fact that the authors raise explicitly these counter-intuitive observations. Given that the authors studied a limited number of clones, and in the absence of rescue experiments, it seems to me that we cannot rule out the possibility that some of the changes observed might be due to clonal effects (“compensatory pathways”) and/or off-target effects introduced by the CRISPR-Cas9 method (“undesired cleavages”). In this framework, what was the rationale behind the authors' decision to choose the heterozygous SNORD115 clone over the homozygous one?

The PWS-AS regions on the maternal and paternal genomic copies of chromosomes 15 undergo distinct methylation and regulation of expression. This may result in unexpected differences between them e.g. distinct wider chromosomal structure and TAD formation (Richer *et al*, 2023). Our aim from the beginning of this study was to limit the genetic modification to the minimum, i.e. using heterozygotes. These also reflect better the situation in patients (human) which, to our knowledge, always have at least one intact copy of Chr15. Initially, due to the lack of second heterozygous H115 deletion mutant we decided to include homozygous cell line in the analysis. However, as mentioned in the manuscript, due to the difference between homozygous and the remaining 3 heterozygous deletion mutants in the RNAseq data we excluded it from further studies. In the interests of space, we have omitted the homozygous line from the revised text.

Regarding the other potential sources of unspecific effects, (1) compensatory pathways: we cannot probably exclude these, but if such effects arise from two different deletions (within the same gene but different transcript) we can envisage that similar events might occur at the organism level; (2) off-target effects: these appear much less likely as we used two different sets of CRIPR-Cas9 guides, and similar off-target effects seem improbable. Moreover, the 2 independent SNORD116 lines show very similar phenotypes. We point this out in the revised MS (P9).

Importantly, do WT cells correspond to cells that have (i) been transfected; (ii) been isolated by cell sorting and (iii) undergone clonal selection, but do not bear a deletion (at least on the paternal allele). Unless I am mistaken, this is not specified in the manuscript; and given the concerns raised above, I think it's very important to avoid the detection of changes that are not directly related to the targeted deletion.

The referee makes a very good point, and we recognize that using a mock-treated control cell line might have been a better choice. WT cells that serve as our negative control represent a bulk population of the parental cell line that was used for CRISPR edition and selection of H115 and H116 clones. This population was not nucleofected or clonally selected. This has frequently been done in the literature, but we cannot exclude the possibility of effects connected with clonal selection. We state this in the revised Supplementary Method section.

However, we note that the SNORD116 cluster deletion and not SNORD115 results in PWS, and the deletions strains have been treated identically. We have therefore focused the revised paper more clearly on effects that are not shared between the cell lines. This has greatly reduced the numbers of significantly altered, potential target genes and we think that this gives improved focus to the results and conclusions.

2-Differentiating LUHMES cells express a homologue of core snoRNP protein Fibrillarin.

That FBLL1 gene is only expressed in brain is very intriguing and unexpected. However, I am not sure what point there is in mentioning it in this manuscript, since the authors indicate that they wish to develop this aspect in another study. In this context, the reader might think that SNORD115 and SNORD116 are preferentially associated with FBLL1 in brain, which to my knowledge has never been demonstrated. Moreover, a role in stabilizing SNORD116 seems unlikely since SNORD116 is expressed in all human tissues analyzed to date (n= 20 human tissues, see Vitali et al 2009 (PMID:20016068)). I propose that the authors withdraw this section, which, in the absence of strong experimental data, does not provide any relevant information as to the regulatory function of SNORD116 or SNORD115.

We feel that these observations do not weaken the MS and would like to retain them here. While this paper was in preparation, we learned that another group has noticed the existence of the *FBLL1* gene. This was presented in a conference poster, but we assume that a publication is in prospect. We would therefore like to record our findings in this MS.

3- Transcriptome and Proteome Analyses

As I understand it, the authors did not attempt to further validate the mis-expression of some

differentially-expressed genes by another technique (RT-qPCR) and using other RNA samples. It is a pity because it could have made the observations more robust and therefore more convincing.

As suggested by the reviewer we supported our RNAseq data by RT-PCR tests performed on the samples used for the generation of RNA-seq libraries and independent samples for direct comparison. Now included in Figure S1C.

Numerous publications have reported the transcriptional and/or post-transcriptional consequences of a gain- or loss-of-function of SNORD116 or SNORD115 genes, or even proposed direct RNA targets, whether in human cell models or in mouse tissues. Among these publications, we can mention Ding et al (2010) PMID: 20195375, Pace et al (2020) PMID: 32365348, Burnet et al (2017) PMID: 27941249, Bochukova et al (2018) PMID: 29590610, Hebras et al (2020) PMID: 33016258, Baldini et al (2022) PMID: 34893870, Falaleev et al (2015) PMID: 26220404; Powel et al (2013) PMID: 23771028. I think it would be interesting for the authors to compare their observations with those already available in the literature, perhaps in the form of a table or Venn diagram. Indeed, there is a lot of disparity in the literature, making it difficult to understand the precise role of SNORD115 and SNORD116. I am obviously aware that it is not always easy to compare different biological contexts but it is important so that the reader can form his own opinion of the state of the field and its relative complexity.

Unfortunately, we were unable to find datasets that could be used for direct comparison. There are papers reporting sequencing of the entire hypothalamus: Bochukova et al (2018) reports whole human postmortem hypothalamus; Pace et al (Pace *et al*, 2020) reports whole mouse hypothalamus. However, it was reported that sequencing adult hypothalamus does not reproduce data from iPSC-derived neurons (Bochukova *et al*, 2018). This agrees with the observation that PWS hypothalamic tissue is depleted in neurons (Bochukova *et al.*, 2018; O'Rahilly & Farooqi, 2006)): i.e. high numbers of DEGs are likely to be consequences of differences in cell type populations rather than specific changes in the transcriptome.

Despite these limitations, we compared our data with RNAseq on hypothalamus from PWS patients from Bochukova et al (2018). The number of overlapping genes together with representation factor and statistical significance were calculated as described (<http://nemates.org/MA/progs/representation.stats.html>) and are presented in Table R1

		DEgenes_d15_116_WT	DEgenes_d15_115_WT
		590	294
number of DE genes in Bochukova_RNAseq (filtered for genes in LUHMES) FDR<0.25	2529	105 Representation factor = 1.5 p< 1.675e-05	53 Representation factor = 1.5 p< 0.001
number of DE genes in Bochukova_RNAseq (filtered for genes in LUHMES) FDR<0.05	439	27 Representation factor = 2.2 p< 1.157e-04	18 Representation factor = 3.0 p<4.663e-05
number of quantified genes in LUHMES	21171		

Table R1

The numbers appear significant. However, it was not clear of what use a lists of genes would be to readers. We have therefore included only a brief comment on the processes that are affected (P11). Comparison of iPSCs-derived neurons from PWS patients and healthy controls (Burnett *et al*, 2017) identified similar terms associated with differentially expressed genes – glycoproteins, extracellular region, receptor signaling. Bochukova *et al* (2018) from PWS brain similarly identify multiple terms associated with synaptic transmission, secretion, ion transporters (we do not observe increase in genes associated with inflammation – we have only neuronal population).

We include further references but have limited these to reports on human cells. The published data shows limited consistency, even for human samples, and we feel that a comprehensive analysis might be better suited to a review article. We also now cite a recent review (Bochukova, 2021).

Finally, it is not clearly stated whether changes in gene expression, as judged by RNA-seq or proteomics, alter the morphology of differentiated cells and/or their behaviour (eg, neuronal activity). In my view, it would be informative to describe the cellular consequences, if any, of the molecular alterations described.

By microscopic inspection of the deletion clones, we saw no obvious differences in morphology and we now point this out (P9). This might be expected, brain development in humans and mice largely proceeds without major problems.

4- Regulation of SNORD116 and SNORD115 synthesis during neuronal differentiation.

Page 7 - “We therefore predict that the apparent extension of the SNHG14 transcripts into the region surrounding the SNORD115 cluster reflects regulated read-through of a termination site located 3’to SNORD116”.

The presence of a transcriptional stop signal positioned between SNORD116 and snoRD115 gene arrays was already predicted in Vitali *et al* 2009 (PMID: 20016068). That the spatiotemporal expression profiles of SNORD115 and SNORD116 do not perfectly overlap during *in vitro* neuronal differentiation or brain development was also described in Vitali *et al* 2009 (PMID: 20016068) and Landers *et al* 2004 (PMID: 15226413). In human, SNORD116 and IPW genes are ubiquitously-expressed while SNORD115 gene expression is brain-specific (neuronal expression), as shown in Vitali *et al* 2009 (PMID: 20016068). More importantly, it has been recently shown that a boundary element overlapping IPW region prevents, in a tissue-specific manner, the expression of SNORD115 and UBE3A-ATS in non-neuronal cells (Martins-Taylor *et al* (2014) PMID: 24363065; Hsiao *et al* 2019 (PMID: 30674673). The above-mentioned references must be quoted and discussed accordingly.

Additional references have been included and are discussed in the revised text.

Page 7 – ‘We conclude that the region encoding SNORD116 is well transcribed in undifferentiated cells. The accumulation of exon regions relative to introns indicates that splicing is functional, suggesting that the failure in mature SNORD116 accumulation reflects instability’.

I find this observation interesting but without any further experimental proofs, it seems to me highly speculative and not so convincing. Indeed, the authors demonstrate the expression of the SNORD116 gene array by sensitive method (RNA-seq) while the expression of the fully processed SNORD116 is monitored by Northern blot. In other words, failure to detect SNORD116 in non-differentiated cells does not mean that it is not expressed. Moreover, we have no idea of the precursor-product relationship, then we can easily imagine a time lag between the detection of the primary transcription product and the nucleolar accumulation of SNORD116 RNP particles. In other words, there is no need to invoke post-transcriptional regulation. Do small RNA-seq data exist that would allow a more relevant and direct comparison? Can the authors detect SNORD116 containing sno-lncRNAs by Northern blot analyses?

We have reworded this section to try to make the observations and conclusions clearer. SNORD116 expression is not absent from non-differentiated cells, and it is readily detected by northern hybridization. However, its abundance relative to total RNA increases substantially during differentiation. In contrast, the abundance of correctly spliced exons, which are derived from the same primary transcript, is not clearly altered. The intronic sequences (other than the mature snoRNAs) are detected at much lower levels than the exons (Fig. 2B), strongly supporting the conclusion that splicing is occurring.

To further address the relation between various SNORD116 containing species: SNORD116 snoRNAs, sno-lncRNAs, and unprocessed introns that could serve a potential “storage” for transcribed but not matured snoRNAs, we performed northern blot hybridization. This did not detect any extended unprocessed transcripts that could suggest delayed maturation (Fig. R2).

Northern analyses with specific probes directed against sno-lncRNAs confirmed that, as predicted from RNAseq, sno-lncRNA4 showed stable expression upon differentiation) and sno-lncRNA1 decreased upon differentiation. The alignment of these observations with RNAseq data, validates use of northern data for approximate quantitation. All the data supports very high expression of the snoRNAs relative to sno-lncRNAs. We include this analysis in the revised manuscript as a Supplementary Figure S1C.

Figure R2 Northern blot analysis of various ncRNAs originating from SNHG14 gene.

Minor comments

Page 3 – “The smallest remove a ~70 Kb region of SNHG14 (snoRNA host gene 14), in which 29 tandem introns each encode the small nucleolar RNA (snoRNA) SNORD116 (Fig. 1)”. In most microdeletions described so far, SNORD109A et the poorly characterized IPW ncRNA are also lacking.

This has been corrected.

Page 3 – “SNHG14 generates a very long non-protein coding RNA (lncRNA) with a predicted primary transcript around 600 Kb in length including 145 annotated introns”. I recommend that the authors quote the original publication, namely Runte et al (2001) PMID: 11726556

Reference has been added (P3).

Page 4 – “Box C/D snoRNAs generally form extended base-paired interactions that precisely target the nucleotide located 4 base-pairs from the box D motif in the snoRNA”. This sentence is very ambiguous. Ribose methylation is targeted at the 2'-hydroxyl of the nucleotide base paired to the 5th, and not the fourth, nucleotide upstream of the D or D'.

This was a typo – many thanks for pointing it out.

Page 5 – “These could not be linked to direct snoRNA base-pairing”.

Have the authors systematically tried to search for potential base-pairings between SNORD115 and SNORD116 and transcripts whose abundance is altered in KO cells?

Unfortunately, current prediction tools return large numbers of hits with high rates of false positives. We approached this using PLEXY, focusing on the smaller number of mRNAs showing differential expression between the line lacking SNORD115 or SNORD116. As expected, we were able to identify several hundred potential regions of complementarity and this is mentioned in the revised text. In the revised text, Supplementary table S7 lists the 631 most stable predicted interactions.

Prediction tools could be supported by the information on the interacting regions between snoRNA and mRNA or the nature of the interaction, which we currently lack. However, we also note that for the acetylation-guide box C/D snoRNAs, the features that define the functional interaction remain unclear – even though the precise modification sites and cognate snoRNAs are known. We hope to address this question using COMRADES, but this is beyond the scope of the current MS.

Page 6 – “We tested snoRNA expression over differentiation time course up to day 15, after which LUHMES cells become sensitive to detaching from the dish”.

What does it mean? Do cells die after 15 days of neuronal differentiation? I think it's important to clarify this sentence because most of the changes observed between KO and WT cells appear late in the course of neuronal differentiation.

Some level of cell death was observed throughout neuronal differentiation. However, between D10 and D15 the majority of cells apparently remained viable and gene expression appeared robust; there were almost no clear changes in mRNA levels in the wildtype. During long term differentiation, all LUHMES cells form a film that loses attachment to the dish and is very sensitive to medium changes. Once they start detaching, the cellular monolayer frequently becomes folded. We discarded these neuronal cultures.

Page 9 - “The maternal chromosome is transcribed on the opposite strand and was left intact”?

What does it mean “opposite strand”? I assume the authors are referring here to the expression of the UBE3A gene on the maternal allele, and not that of SNORD115 and SNORD116 clusters as a non-expert reader might understand. I think it's necessary to clarify this sentence to avoid any confusion.

We have reworded this sentence, hopefully making it clearer (P9). Transcription is in the opposite direction to *SNHG14* and on the other (maternal) chromosome.

Page 19 – “Moreover, mRNAs that are specifically affected show enrichment for neuronal function, including phenotypes related to features of PWS”.

I recommend that the authors mention a few examples.

These have been added.

Page 20 – “The effects on gene expression reported here are much more marked than in several previous analyses”. To which studies do the authors refer? References must be

added

20 – “Previous mouse models for PWS have not fully recapitulated the human disease phenotype”. To which studies do the authors refer? References must be added.

There is a comprehensive review of the PWS mouse models by Bervini et al. from 2013 (Bervini & Herzog, 2013). It is not up to date but clearly shows that deletions introduced in mice only partially recapitulate PWS phenotypes seen in humans. We added a reference to this review in the manuscript.

A targeted deletion of the SNORD116 cluster in SH-SY5Y neuroblastoma human cell line was already reported. The authors must quote explicitly this work and discuss it accordingly.

The paper was referenced and we now explicitly mention SH-SY5Y (P4).

In my view, the authors should quote some references in the last paragraph of the discussion section.

The Discussion has been substantially altered and we have included additional references. References relevant to the SNORD116-specific genes have been placed in Supplementary Table S8. Including a listing all genes and the relevant references in the main text seemed unhelpful.

Although the information is available in the supplementary data, I think it is informative to indicate the number of biological replicates analyzed in the full manuscript.

The MS is somewhat in excess of the maximum word count, so we have just made it clear that the number of replicates is included in SI.

Reviewer #2 (Remarks to the Author):

This was an interesting and potentially important study designed to investigate the impact of paternal loss of the SNORD116 and SNORD115 snoRNA clusters during differentiation of human LUHMES cells that can be differentiated into dopaminergic neurons. The authors describe a progressive snoRNA accumulation, describe a pseudogene transcript of fibularin (FBLL1) whose transcription is anti-correlated with the loss of FBL (involved in C/D snoRNA packaging) during neuronal differentiation. Using CRISPR/Cas9 generated heterozygous deletion models in LUHMES, they further describe changes to the transcriptome and proteome as a result of differentiation and genotypes. Overall, while there are some novel results and some convincing parts (like the deletion models), there is an overall lack of statistical rigor, lack of important controls, and the major conclusions are overstated.

Major concerns:

1. Carrying out differentiated LUHMES neuronal cultures beyond 1 week in culture raises concerns that some of the changes in transcriptome and proteome may be a result of diminished cell viability rather than differentiation. Measurements of live cell counts, reactive oxygen species (ROS) and/or apoptosis should be included for the experiments.

LUHMES cells were used before in the long term experiments. Neuhof et al. characterized electric properties of acid sensing ion channels of LUHMES cells up to day 16 (Neuhof *et al*, 2021); Loser et al. (Loser *et al*, 2021) presented RNAseq data until day 11 of differentiation, and automated patch clamp recordings from day 9. Grams et al (Grams *et al*, 2023) infected LUHMES differentiated for 13 days with HSV-1 and studied the effect of infection until day 21.

As noted above, between D10 and D15 the majority of cells apparently remained viable and gene expression appeared robust; there were almost no clear changes in mRNA levels in the wildtype. To further address the reviewer's concern, we specifically searched for indicators of cell death in the RNA-seq data: (1) We checked for cell death-related GO terms. Autophagy related terms were associated with genes belonging to cluster 0 i.e. not changing during our long-term differentiation (Supplementary table 2). Analysis of H116 DEGs did not reveal any terms containing, apoptosis, apoptotic, death or autophagy. (2) We checked for the expression of death associated genes during the differentiation process. From both of those analysis we conclude that our data are most probably not significantly biased by the diminished cell viability.

Figure R3 Expression of cell death markers in the differentiating wild type and mutant cells.

2. The DEG analysis was performed using EdgeR, which is based on the statistical assumption that most genes in the genomes are unchanged, which is likely not the case for either the differentiation time course or the effects of H116 and H115 mutations. These DEG analyses should be repeated with a more flexible model, as in limma+voom.

In our manuscript we make two kinds of comparisons. First, (1) between the differentiation stages; second (2), between mutant and wild type cells within differentiation timepoints. The first comparison (1) involves comparing undifferentiated cells and day 6 neurons. These are highly different with almost half of the genes identified as differentially expressed. This level of changes to the transcriptome does not fulfil TMM normalization requirements (in EdgeR) for most genes to not change expression. We are aware that statistical power in this case is low. However, we would point out that the only conclusion that we draw from this analysis is that very large number of genes change expression at the beginning of differentiation, and that the differences between d06, d10 and d15 cells are much smaller. We believe that we are authorized to make such a broad statement. For the second comparison (2), global changes between WT and mutant cells within differentiation stages are quite small, as indicated by high correlation values (figure below) or high visual similarity on heatmap representing total transcriptome (Fig. S2B.) There are no tests available to rigorously decide whether samples are similar enough or too distant to be compared with EdgeR. However,

we are confident that samples of analysis (2) are sufficiently similar to be analyzed by EdgeR.

Figure R4 Pearson correlation of all RNAseq samples.

Irrespective of our own judgements, we decided to follow the advice of the reviewer and preform VOOM+LIMMA analysis of the same dataset. We have compared neurons on day 15 when the differences between WT and mutant cells are the highest. We have applied two statistical tests: (1) eBayes, followed by the selection of genes passing the threshold of $LFC = \log_2(2)$ typical for this kind of analysis; and (2) treat that resembles glmTreat used in our original analysis that is more rigorous and allows for lower $LFC = \log_2(1.5)$ threshold.

	DEPs H115 vs WT	DEPs H116 vs WT	H115 vs WT UP	H115 vs WT DOWN	H116 vs WT UP	H116 vs WT DOWN
EdgeR + glmTreat ($LFC = \log_2(1.5)$, $FDR < 0.05$)	292	562	57	235	95	467
VOOM + Limma + eBayes ($LFC = \log_2(2)$, $FDR < 0.05$)	522	1359	145	377	192	1167
VOOM + Limma + treat ($LFC = \log_2(1.5)$, $FDR < 0.05$)	109	346	15	94	26	320

Table R2

The overlap between the differentially genes from our initial edgeR analysis and more flexible VOOM + limma approach suggested by the reviewer is very high, which strongly suggests that the use of EdgeR in the case of our analysis was justified. In such a case we would like to keep the original analysis in the publication.

Figure R5. Overlap between the results of differential expression analysis using EdgeR and VOOM+LIMMA (VL)

3. The k-means clustering approach is somewhat rudimentary way of clustering genes by coexpression. A Weighted Gene Co-expression Network Analysis (WGCNA) approach would be better for finding groups of genes that change over time and by genotype. (Langfelder, P. and Horvath, S., 2008).

The aim of our analysis was to look globally on the expression of genes upon differentiation, to observe when most of the changes happen and how much time it takes for the transcriptome to stabilize. Initially we chose the k-means clustering for its simplicity. We tested various values of k, and selected visually the one which gave the best separation between clusters. Now, following the advice from the reviewer we performed clustering analysis using WGCNA. This is a very robust method for identifying coherent groups of genes within large datasets that associate with specific phenotypes, identification of networks and key genes.

We used all default settings except minModuleSize=100, to decrease the detail of the analysis. Consequently, we ended up with 8 modules/clusters with following expression pattern.

GO:BP	Term Name	Term ID	p_adj	p_adj	p_adj	p_adj
	nervous system development	GO:0007399	4.221x10 ⁻⁶	7.620x10 ⁻⁹	4.624x10 ⁻²⁹	2.265x10 ⁻⁶²
	system development	GO:0048731	1.209x10 ⁻¹²	1.693x10 ⁻¹⁶	6.676x10 ⁻²⁷	1.730x10 ⁻⁶¹
	multicellular organism development	GO:0007275	3.460x10 ⁻¹¹	5.159x10 ⁻¹⁴	1.219x10 ⁻²⁷	3.885x10 ⁻⁵²
	synaptic signaling	GO:0099536	4.000	6.067x10 ⁻¹	2.613x10 ⁻¹⁷	1.433x10 ⁻⁴⁸
	anterograde trans-synaptic signaling	GO:0098916	4.000	8.316x10 ⁻¹	1.981x10 ⁻¹⁶	1.980x10 ⁻⁴⁸
	chemical synaptic transmission	GO:0007268	4.000	8.316x10 ⁻¹	1.981x10 ⁻¹⁶	1.980x10 ⁻⁴⁸
	trans-synaptic signaling	GO:0099537	4.000	9.736x10 ⁻¹	3.771x10 ⁻¹⁶	6.466x10 ⁻⁴⁸
	neuron development	GO:0048666	1.436x10 ⁻²	2.881x10 ⁻⁶	3.484x10 ⁻²⁵	2.150x10 ⁻⁴³
	neuron projection development	GO:0031175	1.072x10 ⁻²	8.829x10 ⁻⁶	8.743x10 ⁻²⁵	4.727x10 ⁻⁴³
	neurogenesis	GO:0022008	1.079x10 ⁻⁵	2.548x10 ⁻⁷	1.370x10 ⁻²¹	1.507x10 ⁻⁴²
	generation of neurons	GO:0048699	1.149x10 ⁻³	3.543x10 ⁻⁸	1.285x10 ⁻²³	2.641x10 ⁻⁴²
	plasma membrane bounded cell projection organi...	GO:0120036	3.046x10 ⁻²	4.704x10 ⁻⁴	1.980x10 ⁻³¹	1.560x10 ⁻⁴¹
	cell projection organization	GO:0030030	7.278x10 ⁻³	1.791x10 ⁻⁴	1.591x10 ⁻³¹	4.370x10 ⁻⁴¹
	neuron differentiation	GO:0030182	2.244x10 ⁻⁴	5.468x10 ⁻⁸	2.971x10 ⁻²⁴	1.039x10 ⁻⁴⁰
	cell-cell signaling	GO:0007267	5.621x10 ⁻⁴	4.066x10 ⁻⁵	1.274x10 ⁻¹⁶	8.910x10 ⁻⁴⁰
	neuron projection morphogenesis	GO:0048812	4.000	4.000	1.259x10 ⁻¹⁹	1.899x10 ⁻³⁸
	cell projection morphogenesis	GO:0048858	4.000	4.000	5.620x10 ⁻¹⁹	2.067x10 ⁻³⁷
	plasma membrane bounded cell projection morpho...	GO:0120039	4.000	4.000	9.786x10 ⁻¹⁹	3.801x10 ⁻³⁷
	modulation of chemical synaptic transmission	GO:0050804	4.000	4.000	4.321x10 ⁻¹³	4.301x10 ⁻³⁷
	regulation of trans-synaptic signaling	GO:0099177	4.000	4.000	4.883x10 ⁻¹³	5.359x10 ⁻³⁷
	cell part morphogenesis	GO:0032990	4.000	4.000	6.557x10 ⁻¹⁸	7.517x10 ⁻³⁶
	cell morphogenesis	GO:0000902	4.000	6.949x10 ⁻¹	1.267x10 ⁻¹⁶	1.472x10 ⁻³⁴
	cellular component morphogenesis	GO:0032989	4.000	4.000	6.209x10 ⁻¹⁴	3.759x10 ⁻³²
	cell morphogenesis involved in neuron differentia...	GO:0048667	4.000	4.000	1.933x10 ⁻¹⁹	4.830x10 ⁻³²
	anatomical structure morphogenesis	GO:0009653	6.572x10 ⁻⁸	1.338x10 ⁻⁹	1.136x10 ⁻¹⁵	1.140x10 ⁻³¹
	cell junction organization	GO:0034330	4.000	6.298x10 ⁻¹	3.812x10 ⁻¹⁵	5.015x10 ⁻³¹
	synapse organization	GO:0050808	4.000	4.000	4.015x10 ⁻¹³	6.551x10 ⁻³¹
	axon development	GO:0061564	4.000	9.487x10 ⁻¹	8.457x10 ⁻¹⁴	1.733x10 ⁻²⁹
	regulation of biological quality	GO:0065008	1.991x10 ⁻⁶	9.593x10 ⁻⁸	7.140x10 ⁻¹⁰	5.086x10 ⁻²⁹
	cell development	GO:0048468	2.753x10 ⁻⁷	1.246x10 ⁻⁸	4.795x10 ⁻¹⁵	1.007x10 ⁻²⁷
	cell differentiation	GO:0030154	3.109x10 ⁻¹¹	1.837x10 ⁻¹¹	4.660x10 ⁻¹¹	2.636x10 ⁻²⁷
	cellular developmental process	GO:0048869	3.189x10 ⁻¹¹	1.884x10 ⁻¹¹	4.831x10 ⁻¹¹	2.782x10 ⁻²⁷
	axonogenesis	GO:0007409	4.000	4.000	1.601x10 ⁻¹³	4.452x10 ⁻²⁷
	cell population proliferation	GO:0008283	8.785x10 ⁻⁹	3.014x10 ⁻⁸	8.192x10 ⁻¹⁸	5.021x10 ⁻²⁷
	regulation of transport	GO:0051049	2.432x10 ⁻¹	4.787x10 ⁻³	7.022x10 ⁻¹¹	5.581x10 ⁻²⁶
	regulation of membrane potential	GO:0042391	4.000	4.000	1.348x10 ⁻³	3.870x10 ⁻²⁵
	synapse assembly	GO:0007416	4.000	4.000	6.691x10 ⁻¹²	6.467x10 ⁻²⁵
	regulation of localization	GO:0032879	8.453x10 ⁻³	3.877x10 ⁻³	3.511x10 ⁻¹¹	2.453x10 ⁻²³
	cell junction assembly	GO:0034329	4.000	4.000	1.347x10 ⁻¹⁰	1.179x10 ⁻²²
	regulation of cell communication	GO:0010646	2.126x10 ⁻¹³	2.013x10 ⁻⁷	2.661x10 ⁻¹⁰	2.730x10 ⁻²²
	cell adhesion	GO:0007155	6.224x10 ⁻¹	4.082x10 ⁻⁶	2.188x10 ⁻⁵	1.571x10 ⁻²⁰
	regulation of signaling	GO:0023051	1.609x10 ⁻¹³	5.719x10 ⁻⁷	3.177x10 ⁻¹⁰	2.467x10 ⁻²⁰
	regulation of neuron projection development	GO:0010975	4.000	4.000	3.106x10 ⁻¹⁶	5.019x10 ⁻²⁰
	behavior	GO:0007610	4.000	6.064x10 ⁻⁴	1.479x10 ⁻⁹	7.358x10 ⁻²⁰
	neuron projection guidance	GO:0097485	4.000	4.000	2.619x10 ⁻⁸	1.027x10 ⁻¹⁸
	axon guidance	GO:0007411	4.000	4.000	2.619x10 ⁻⁸	1.027x10 ⁻¹⁸
	regulation of synapse structure or activity	GO:0050803	4.000	4.000	7.734x10 ⁻¹²	1.495x10 ⁻¹⁸
	regulation of plasma membrane bounded cell proj...	GO:0120035	4.000	4.000	1.992x10 ⁻¹⁵	6.666x10 ⁻¹⁸
	regulation of developmental process	GO:0050793	8.564x10 ⁻⁶	1.733x10 ⁻⁸	3.053x10 ⁻¹¹	2.555x10 ⁻¹⁷
	regulation of cellular component organization	GO:0051128	1.149x10 ⁻³	2.016x10 ⁻¹	9.431x10 ⁻¹³	3.409x10 ⁻¹⁷
	regulation of cell projection organization	GO:0031344	4.000	4.000	1.199x10 ⁻¹⁵	4.029x10 ⁻¹⁷

Figure R6
Comparison between the results of k-means clustering and WGCNA

Similarly to our previous analysis WGCNA identifies clusters of genes that change late in differentiation and correspond to previously identified clusters CL4 (module brown) and CL5 (modules red and yellow)

and yellow) that were of our main interest. The overlap between genes belonging to k-means clusters and corresponding WGCNA modules is moderate with approximately 50% of genes shared. We used the lists of genes in CL4, CL5, brown and red/yellow clusters in the multiquery analysis by gProfiler (on the left). Very clearly biological processes associated with yellow/red module align with those associated with CL5 and brown with CL4.

Although we appreciate the robustness of the WGCNA analysis it was not evident that this method gave greater insight into our analysis and believe that k-means is adequate for our simple task. We would like to maintain our initial analysis in the publication.

4. The statement “Fig. S3). Comparison of D00 with D10 showed some reduction in pre-rRNA abundance during differentiation, consistent with the exit from cell division.” Was not supported by any quantification or statistics. These results are not apparent by looking at the blots.

This statement was based on the visual inspection. We now simply state that no strong changes were observed.

5. There are also no statistics to back up the conclusions being made about transcript and protein levels in Fig 5 C-E.

We have added statistical significance indicators to the plots.

6. The results in Fig 6D-F, showing an increase in number of DEGs in H116 and H115 compared to WT at D10 and D15, could be the result of a reduction in cellular health and viability in these cultures rather than differences in differentiation, as the authors conclude. Cell stress in the cultures is implicated by the vast reduction in transcript levels compared to WT LUHMES at the same time points.

There were no clear indications of altered cellular health or viability in the H116 or H115 cell lines compared to WT, at D10 or D15. We also saw no indication of gross changes in neuronal morphology or differentiation – e.g. axon outgrowth or neurite formation.

A large number of genes showed altered expression during differentiation in WT cells, but these predominantly showed the same changes in the mutants, with only a relatively small number of differences. We have altered the text to try make this clearer (P10).

Our conclusion regarding the differences in timing of differentiation is based on the comparison of our data with single-cell sequencing from differentiating midbrain from Mano *et al.* However, this kind of analysis is always focused on a limited number of genes characteristic for specific cell type i.e. differentiation stage, and the reduction in cellular health and viability in mutant cells cannot be excluded, and may even be likely. Based on microscopy, Bochukova *et al.*, reported that a deletion encompassing the SNORD116 cluster in SH-SY5Y cells leads to reduced neuronal differentiation, proliferation, and survival. Our “progressed differentiation” may mean faster escape from the cell cycle, shorter expression of genes responsible for neurite growth or synaptic development, so there is no contradiction here. Reduction in the number of neurons in some areas of PWS brains including hypothalamus was previously observed (Manning & Holland, 2015) which is possibly neurodevelopmental (Brown *et al.*, 2022).

7. This unexpected result stated in the Results is also a concern: “Intriguingly, some changes in mRNA accumulation were detected in undifferentiated cells lacking SNORD115 or SNORD116 clusters (106 in H116 and 38 in H115, 21 common). This was unexpected,

especially for H115 cell line, as undifferentiated LUHMES cells apparently lack transcription across the SNORD115 cluster.” An alternative explanation for this phenomenon is that some of the DEGs are due to inter-individual differences in gene expression between different cell lines grown in separate flasks rather than SNORD-dependent. To address this, the authors should examine a comparison of separate WT LUHMES cell cultures compared to each other for a background level of DEGs seen independently of genotype differences. Since the genotype differences observed are quite subtle in this study, this WT-WT control is especially important to rule out possible background noise.

Our analysis is performed using biological triplicates, samples were grown independently at different time. Based on the correlations (see response above), hierarchical clustering and PCA analysis we clearly see high similarity between WT cell replicates. In addition, the RNAseq analysis methods that we used including EdgeR (with correction for batch effects), consider the variation within replicates and between different samples. Consequently, genes with higher than expected variability that cannot be explained by the differences in genotype do not pass statistical testing and are not reported as DEGs. Additional WT-WT comparisons should therefore not be needed.

8. This major conclusion is overstated, based on the limited data shown in support and lack of cell viability controls: “We conclude that the data support a model that the loss of SNORD116 advances developmental timing.”(Discussion) and “Most changes in mRNA and protein abundance appeared relatively late in development...”(Abstract) Comparisons to in vivo fetal brain transcriptome and transcription factor maps are only correlative and the authors have not ruled out alternative interpretations of the results.

As noted above, the data give no indication of problems with cell viability and we saw no signs of defects in gross neuronal morphology. The suggested alteration in development timing was explicitly presented as a model to explain the results obtained. There is no doubt that alternative explanations could be devised, but this appeared to best fit the available data. In the revised MS these sections have been rewritten, as we refocused the MS on the differences between the effects of loss of SNORD116 and SNORD115, as described above.

Reviewer #3 (Remarks to the Author):

In “Roles of SNORD115 and SNORD116 ncRNA clusters in neuronal differentiation” Helwak and colleagues report a comprehensive survey of changes triggered by differentiation in a neuronal model. Extensive comparisons are made between wild-type and mutant cell lines generated with genome editing. Minimal alterations are detected at the level of splicing. Changes in gene expression, post-transcriptional regulation, and transcription are reported. It appears that differentiation is more rapid in the mutant cell lines based on inference from molecular measurements. There is overlap for some of the targets suggesting a functional interaction between the snoRNAs. Intriguingly, loss of the snoRNAs results in increased activity of several TFs. This work is comprehensive and technically sound. There are limitations regarding lack of mechanistic insights into precisely how the snoRNAs function. However, the manuscript is well written and clear. This work will likely be of broad interest to the field. I have several comments and questions rooted in genuine curiosity that I suspect

will be shared by other readers. I hope that addressing these comments will result in an improved manuscript:

1. Given the similarities between H116 or H115, can the authors comment on what the consequence would be from a double mutant? Does this happen in any clinically relevant instances and is the outcome more deleterious than the SNORD116 deletion? Perhaps this would have a stronger phenotype that would be useful for future mechanistic investigations.

We have not assessed this point. Most clinical deletions take out a much larger region than SNORD116, including SNORD115 cluster; and most of the genes in PWS region have a potential to contribute to the phenotype. But the microdeletion of SNORD116 is reported to largely recapitulate the features of the canonical deletions.

2. How do SNORD115/116 influence transcription factor activity? Do the authors think this is a direct or indirect effect? The authors rule out many of the obvious explanations that could explain this effect so what options remain? While clearly outside the scope of this manuscript, it would be of interest to identify the RNA/protein targets/partners of SNORD115/116. This work provides an essential resource for gauging the best timepoint at which to attempt this key experiment and to provide essential negative control cell lines.

There have been several recent reports of functional interactions between transcription factors and ncRNAs. For review see (Marchese *et al*, 2017).

We intend to characterize the RNA and protein interaction partners for SNORD116 during differentiation. However, this will represent a major body of work that we feel lies beyond the scope of the present publication.

3. How plastic are the effects of SNORD115/116 loss? Would expression late in development reverse the effects on transcription factor activity? Can the authors comment on the temporal dynamics of SNORD115/116 function? Is there any evidence that would enable discussion of this point as it has implications for potential salubrious use of this information for Prader-Willi syndrome.

This is indeed a key question that we hope to address. The outcome would inform potential future therapeutic interventions.

4. The FBLL1 connection is intriguing. What are the physiologic consequences of FBLL1 elimination? Do these phenocopy loss of SNORD115/116? If this is the basis of a separate study, that's fine but I think it would be worth noting if there are existing data on this point.

There are no published functional data on FBLL1. We are in the process of generating such data, but feel that they lie beyond the scope of the current MS.

Minor points:

5. Abstract – apparently is used in back to back sentences. Suggest more diverse word choice

The wording has been changed.

6. Page 18 discussion – LUHMES might be too low high. Suggest correcting typo to low.

We have corrected this read:

“... the basal level of expression in LUHMES might be too high....”

7. Fig 8A, esp. right panel: it is quite difficult to find some of the replicates on the proteome PCA plot due to the dots overlapping. The symbols might be made smaller so nearby points do not obscure one another.

We have reduced all points to 70% of the original size.

8. Page 14: typo "...least mature AT the foot and the most mature at the top."

Corrected

References:

- Bervini S, Herzog H (2013) Mouse models of Prader-Willi Syndrome: a systematic review. *Front Neuroendocrinol* 34: 107-119
- Bochukova EG (2021) Transcriptomics of the Prader-Willi syndrome hypothalamus. *Handb Clin Neurol* 181: 369-379
- Bochukova EG, Lawler K, Croizier S, Keogh JM, Patel N, Strohbehn G, Lo KK, Humphrey J, Hokken-Koelega A, Damen L *et al* (2018) A Transcriptomic Signature of the Hypothalamic Response to Fasting and BDNF Deficiency in Prader-Willi Syndrome. *Cell Rep* 22: 3401-3408
- Brown SSG, Manning KE, Fletcher P, Holland A (2022) In vivo neuroimaging evidence of hypothalamic alteration in Prader-Willi syndrome. *Brain Communications* 4
- Burnett LC, LeDuc CA, Sulsona CR, Paull D, Rausch R, Eddiry S, Carli JF, Morabito MV, Skowronski AA, Hubner G *et al* (2017) Deficiency in prohormone convertase PC1 impairs prohormone processing in Prader-Willi syndrome. *J Clin Invest* 127: 293-305
- Grams TR, Edwards TG, Bloom DC (2023) A viral IncRNA tethers HSV-1 genomes at the nuclear periphery to establish viral latency. *J Virol* 97: e0143823
- Loser D, Schaefer J, Danker T, Möller C, Brüll M, Suci I, Ückert AK, Klima S, Leist M, Kraushaar U (2021) Human neuronal signaling and communication assays to assess functional neurotoxicity. *Arch Toxicol* 95: 229-252
- Manning KE, Holland AJ (2015) Puzzle Pieces: Neural Structure and Function in Prader-Willi Syndrome. *Diseases* 3: 382-415
- Marchese FP, Raimondi I, Huarte M (2017) The multidimensional mechanisms of long noncoding RNA function. *Genome biology* 18: 206
- Neuhof A, Tian Y, Reska A, Falkenburger BH, Gründer S (2021) Large Acid-Evoked Currents, Mediated by ASIC1a, Accompany Differentiation in Human Dopaminergic Neurons. *Front Cell Neurosci* 15: 668008
- O'Rahilly S, Farooqi IS (2006) Genetics of obesity. *Philos Trans R Soc Lond B Biol Sci* 361: 1095-1105

Pace M, Colombi I, Falappa M, Freschi A, Bandarabadi M, Armirotti A, Encarnación BM, Adamantidis AR, Amici R, Cerri M *et al* (2020) Loss of Snord116 alters cortical neuronal activity in mice: a pre-clinical investigation of Prader-Willi syndrome. *Hum Mol Genet*

Richer S, Tian Y, Schoenfelder S, Hurst L, Murrell A, Pisignano G (2023) Widespread allele-specific topological domains in the human genome are not confined to imprinted gene clusters. *Genome biology* 24: 40

Shah RR, Cholewa-Waclaw J, Davies FCJ, Paton KM, Chaligne R, Heard E, Abbott CM, Bird AP (2016) Efficient and versatile CRISPR engineering of human neurons in culture to model neurological disorders. *Wellcome Open Res* 1: 13

Response to reviewer comments:

Reviewer #1 (Remarks to the Author):

I am grateful to the authors for their efforts and additional work during the revision process. I am generally satisfied with their responses to my questions and comments. The modifications made to the revised version have improved its readability. If the editorial committee of Nature Communications decides to accept this manuscript for publication, it is essential to include critical elements in the discussion by addressing certain inherent limitations of this type of study.

In the full manuscript, the authors must mention and discuss that WT controls correspond to parental cells that did not undergo the same transfection and clonal selection process as the edited cells. Therefore, we cannot exclude the possibility that some differences between WT and mutated cells are related to other factors associated with the experimental procedure.

We have altered the Results (p9) and Discussion (p17) to make this limitation clearer. This concern does not arise for the comparison of the H115 and H116 lines, as all were derived in parallel.

Additionally, the initial version of the manuscript stated that KO115 homozygotes resembled WT more than heterozygotes. Unless I am mistaken, this counterintuitive information is no longer mentioned in the revised version. For transparency and to allow readers to form their own opinions, it is crucial that this information is mentioned in the full manuscript (Do some of the observed differences genuinely result from the absence of SNORD115 itself?). This could be achieved by adding a single sentence, without needing to show experimental data.

This information was included in the SI of the revised MS, but we have also now included it in the Results (p9).

In my opinion, the part concerning FBLL1 does not integrate well with the rest of the study, especially since the authors do not provide, at the very least, an immunoprecipitation assay indicating that SNORD116 could associate with FBLL1. With the current data, the discussion of a putative role of FBLL1 in phase separation is intriguing but appears too speculative without experimental support. Furthermore, mentioning a link between FBLL1 and the stabilization of SNORD116 in the abstract risks leading to misinterpretation among readers. Nevertheless, I understand the authors' position and leave it to the editorial committee to make a decision on this matter.

As both referees recommend removal of these data, we have done so.

Reviewer #2 (Remarks to the Author):

While I appreciate that the authors attempted to respond to all of the reviewers' concerns, the remaining problem is that very few of these suggestions for improvement were actually incorporated into the revised manuscript. As a result, the revised version of the manuscript is not an improvement. If anything, it is less impactful and more confusing because the authors

have watered down the major conclusions in the abstract (such as the developmental timing one), yet they are still in the Results and Discussion sections. The suggestions for improving the rigor of the experiments through additional methods and approaches were not included in the manuscript. To be more specific, here are the remaining concerns:

Rev1 concerns:

The concerns regarding transcriptome effects that may have resulted from the lack of an appropriate control (mock transfected) and/or clonal and off-target effects were not addressed appropriately and my concern is about reproducibility.

As noted above, we have altered the text to make this limitation more explicit. This concern should not be relevant to the H115 to H116 comparison, since these lines were treated identically.

The reasons for wanting to keep the FBLL1 results in the manuscript were not appropriate. In the revised abstract, this section seems very disjointed from the rest of the manuscript.

We have removed these results.

The request for comparison of DEG results to other studies was not appropriate, as not all studies mention were hypothalamus (Powell et al is mouse cortex) and the overlap can be done at the level of GO terms and KEGG pathways which are less sensitive to bulk tissue concerns. If the authors are claiming that these cell line models are relevant to PWS, they need to provide evidence for that claim.

Following the previous request from the reviewer, we looked for publications suitable for this comparison. We agreed with the conclusion suggested by the reviewer, that the best solution is to compare the GO terms associated with DEGs. Indeed, we observed the overlap with the published data, and we reported this in the revised text. To further emphasize this point we now also mention this observation in the Discussion.

For comparison of developmental timing, the data on these cell lines are taken from gene expression databases cataloguing gene expression during neurodevelopment. They are not related to PWS and we had not intended to make any claim to this effect. These analyses compared the changes in gene expression in the deletion mutant relative to the wildtype, with normal cells at different developmental stages.

Figure R2 was not at all convincing in showing the SNHG14 transcript by Northern blot.

The SNHG14 primary transcript is around 600 Kb in length, includes 145 annotated introns and has more than 100 predicted alternatively spliced isoforms. It cannot be visualized by northern analysis – only short fragments of the nascent or mature transcripts are resolved. The northern blots in Fig S1C were included at the request of the referee, to observe in more detail maturation of snoRNAs and to look for the precursor-product relationship during splicing. They were not intended for use in visualization or quantitation of the SNHG14 transcripts. We have altered the text to try to make this clearer.

Rev 2 concerns:

The concern about cells at d10 and d15 should be able to provide RNA quality scores in the response.

Included in Supplementary Table S1

The reason for suggesting additional methods for DEG analysis (Limma voom) and systems biology (WGCNA) was so that the authors could back up their major conclusion about developmental timing with additional methods. It appears that the authors were unable to back up the developmental timing claim with these methods because they were not included in the revised manuscript.

This point was addressed at length in the response to the referee. We now include mention that these analyses were performed.

The concern about transcriptome results potentially being the result of cellular heterogeneity raised by both reviewers has not been appropriately addressed in the revised manuscript.

As noted above (Ref. 1), we have more explicitly addressed this limitation of the work in the Results and Discussion.

Rev3 concerns:

This reviewer had several suggestions and questions, none of which were incorporated into the revised manuscript.

We note that Referee 3 considered that the concerns raised had been suitably addressed.

Reviewer #3 (Remarks to the Author):

Thank you for addressing my comments.

We are grateful that Referee 3 considered that the concerns raised had been suitably addressed.